# Be Confident in What You Know: Bayesian Parameter Efficient Fine-Tuning of Vision Foundation Models

**Deep Shankar Pandey** [†]   **Spandan Pyakurel** [†]   **Qi Yu**[*]

Rochester Institute of Technology

{dp7972,sp1468,qi.yu}@rit.edu

## Abstract

Large transformer-based foundation models have been commonly used as pre-trained models that can be adapted to different challenging datasets and settings with state-of-the-art generalization performance. Parameter efficient fine-tuning (PEFT) provides promising generalization performance in adaptation while incurring minimum computational overhead. However, adaptation of these foundation models through PEFT leads to accurate but severely underconfident models, especially in few-shot learning settings. Moreover, the adapted models lack accurate fine-grained uncertainty quantification capabilities limiting their broader applicability in critical domains. To fill out this critical gap, we develop a novel lightweight Bayesian Parameter Efficient Fine-Tuning (referred to as `Bayesian-PEFT`) framework for large transformer-based foundation models. The framework integrates state-of-the-art PEFT techniques with two Bayesian components to address the under-confidence issue while ensuring reliable prediction under challenging few-shot settings. The first component performs base rate adjustment to strengthen the prior belief corresponding to the knowledge gained through pre-training, making the model more confident in its predictions; the second component builds an evidential ensemble that leverages belief regularization to ensure diversity among different ensemble components. Our thorough theoretical analysis justifies that the Bayesian components can ensure reliable and accurate few-shot adaptations with well-calibrated uncertainty quantification. Extensive experiments across diverse datasets, few-shot learning scenarios, and multiple PEFT techniques demonstrate the outstanding prediction and calibration performance by `Bayesian-PEFT`.

## 1  Introduction

Transformer-based foundation models have been developed as general models with state-of-the-art generalization performance [32, 66, 52, 28]. These models leverage the rich meta-knowledge acquired during the pre-training stage to effectively adapt to complex downstream tasks [32], where pre-training is usually performed on massive-scale annotated datasets (*e.g.,* [35, 55]) through supervised learning[32, 66, 28] or self-supervised learning [52]. To achieve effective adaption, various parameter-efficient fine-tuning (PEFT) approaches have been developed [38, 27, 9, 54] that introduce a small number of tunable parameters either within or outside of the backbone architecture to ensure good generalization capability while incurring little computational overhead because most parts of (or the entire) backbone architecture is frozen during fine-tuning [25, 59, 67]. Bias-fine tuning [9], a representative partial tuning-based PEFT, only fine-tunes the bias parameters to downstream tasks. Adapter [51] and side-tune [72] fine-tuning techniques are instances of extra module-based PEFT that introduce extra parameterized modules and fine-tune them based on the downstream tasks. Visual Prompt-tuning [32] (VPT) follows the popular prompt learning paradigm by introducing a learnable

---

[*]Corresponding author, [†] equal contribution

38th Conference on Neural Information Processing Systems (NeurIPS 2024).

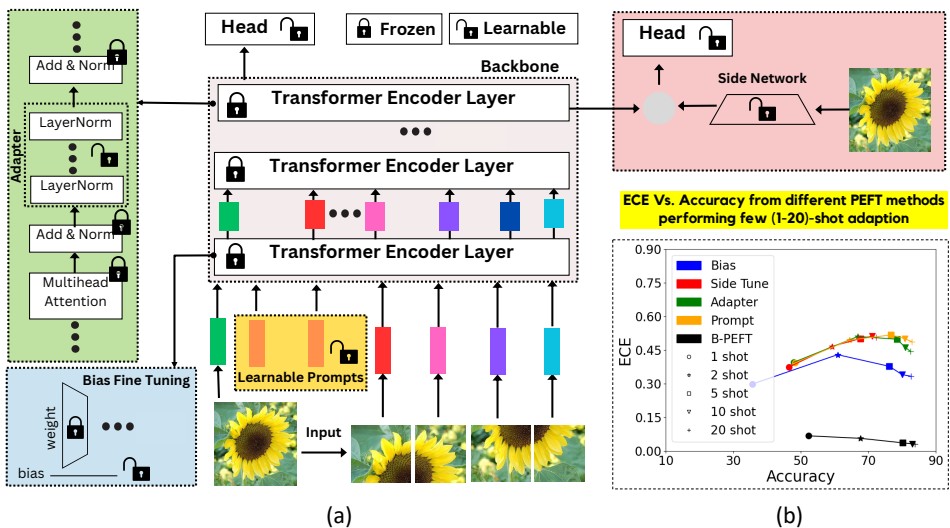

Figure 1: Accuracy Vs. ECE on CIFAR100 few-shot adaptation from different PEFT methods

prompt that is fine-tuned on the downstream task knowledge keeping the pre-trained backbone frozen.

Despite the attractive generalization performance, most foundation models adapted to downstream few-shot tasks through PEFT exhibit a somewhat surprising and undesirable behavior that may prevent them from being applied to many critical domains. Figure 1 (b) shows the predictive accuracy versus the Expected Calibration Error (ECE) of a foundation model after performing few-shot adaptation on CIFAR-100 using a series of representative PEFT methods, including VPT [32], Adapter [51], Bias Fine Tuning [71], and Side-tune [72]. It is clear that the adapted model is able to provide accurate predictions even after fine-tuned on limited training samples. For example, all PEFT methods help to boost the model's accuracy to over 75% using just 5-shot fine-tuning and the accuracy reaches 80% after 10-shot fine-tuning. However, the adapted model is very poorly calibrated as shown by the large ECE scores, which are consistently over 0.3 across all the fine-tuning methods. Increasing the fine-tuning size does not show clear improvement and sometimes even hurts the calibration performance. While one may expect the poor ECE to be caused by over-confidence as we fine-tune a large foundation model using very limited training samples, the detailed ECE plots as shown in Figure 2 (a)-(c) reveal that the model is in fact severely under-confident. For example, after adapting to the 1-shot training dataset, the VPT fine-tuned model can already achieve a test accuracy close to $50\%$ but is under-confident in almost all its predictions leading to an ECE score over $0.45$. The under-confidence issue is observed for all representative PEFT methods across different datasets and various few-shot learning settings as evidenced by our experiments.

The under-confident few-shot adaptation behavior of foundation models closely mimics how human experts with rich domain knowledge in their own disciplines tend to make *conservative* decisions when facing new tasks that deviate from their own expertise. Analogous to their human counterparts, the rich prior knowledge gained through the pre-training stage of foundational models overshadows the relatively limited knowledge obtained through few-shot fine-tuning, which prevents them from making more confident predictions in the downstream tasks. Unreliable uncertainty (*i.e.,* confidence) quantification makes the predictions provided by these models less trustworthy, which may limit applying the promising "pre-train-then-finetune" paradigm to many critical domains. As shown in Figure 2 (a)-(c), the fine-tuned model seldom makes any predictions with confidence over 80%, making it difficult to leverage these predictions in any high-stakes decision-making process.

The need to balance between the rich prior knowledge gained through pre-training and the new knowledge obtained through few-shot adaptation inspires us to investigate the under-confidence issue from the Bayesian perspective. In particular, we propose to look into the predictive behavior of the few-shot adapted foundation model through the lenses of evidential learning [56], which is built upon Bayesian theorem and subjective logic (SL) theory [33]. As part of the recent advances in modern Bayesian modeling, evidential learning provides a cost-effective way to perform Bayesian inference with the capability to quantify fine-grained second-order uncertainty [57]. By leveraging the important

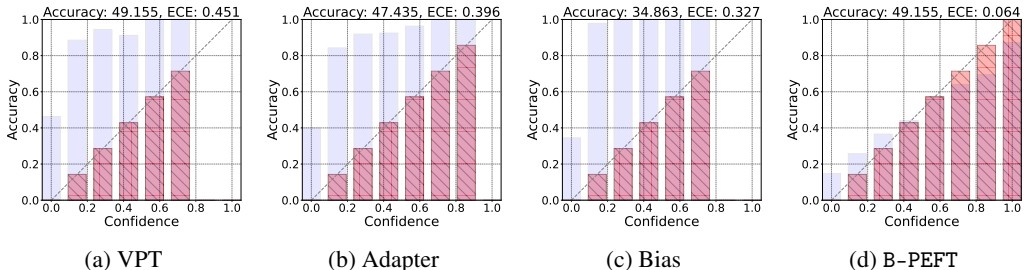

Figure 2: PEFT on the 1-shot CIFAR100 dataset: all existing PEFT techniques exhibit severe under-confidence while the proposed B-PEFT reduces the ECE by almost an order of magnitude.

theoretical connection between fine-grained uncertainty and model accuracy [46] , we unveil the underlying reason that supports the good predictive performance of a few-shot adapted foundation model and the root cause for the under-confident behavior. Drawing from this important insight as outlined above, we propose to integrate the modern PEFT techniques into a novel lightweight Bayesian framework, referred to as Bayesian-PEFT (B-PEFT), aiming to achieve highly reliable and accurate few-shot adaptations with well-calibrated and trustworthy uncertainty quantification.

The proposed Bayesian framework offers two important components to address the under-confidence issue while ensuring reliable prediction under challenging few-shot settings. The first component makes novel adjustments to the base rates introduced by the SL theory to strengthen the prior belief corresponding to the knowledge gained through pre-training. Meanwhile, the adjustment does not change the relative order of the belief assigned to different classes, which ensures that the model accuracy is maintained. Our theoretical analysis shows that the proposed base rate adjustment strategy leads to more confident predictions (by increasing the gaps between the belief assigned to the ground-truth class and the rest) without compromising the model's accuracy. Figure 2d shows that B-PEFT significantly improves the model calibration. To further enhance the reliability of both prediction accuracy and uncertainty quantification when performing few-shot adaptation, the second component performs Bayesian model averaging by building a diversity-inducing evidential ensemble. In addition to using different random initialization of the PEFT components, diversity is further enhanced through incorrect belief regularization that penalizes a model for assigning a high belief to a non-ground-truth class. By controlling the strength of belief regularization, different ensemble components are guided to learn from diverse features in the data space, where a light penalty allows an ensemble component to learn the common discriminative features while a heavy one will force an ensemble component to learn rare features to avoid errors on the difficult data samples. A deeper theoretical analysis of the proposed diversity-induced evidential ensemble is equivalent to Stein Variational Gradient Descent (SVGD) based ensembles [13]. Experiments on multiple challenging few-shot learning tasks justify the superior performance of B-PEFT over state-of-the-art PEFT baselines, in terms of both prediction accuracy and uncertainty calibration performance. Our contributions can be summarized as follows:

- We identify the severe under-confidence issue of pre-trained foundation models after performing parameter-efficient fine-tuning over few-shot datasets. The fine-grained uncertainty analysis through evidential learning and SL theory reveals the root cause for their good predictive performance while being under-confident.
- We develop a novel lightweight Bayesian framework that integrates state-of-the-art PEFT techniques with two Bayesian components to address the under-confidence issue while ensuring reliable prediction under challenging few-shot settings. The first component performs base rate adjustment to strengthen the prior belief corresponding to the knowledge gained through pre-training, making the model more confident in its predictions; the second component builds an evidential ensemble that leverages belief regularization to ensure diversity among different ensemble components.
- We perform thorough theoretical analysis to justify why the proposed Bayesian components can ensure reliable and accurate few-shot adaptations with well-calibrated uncertainty quantification.
- We carry out experiments with 4 benchmark datasets, 5 different few-shot settings, and 3 parameter efficient fine-tuning techniques that demonstrate the effectiveness of the developed model.

## 2 Related Works

**Parameter Efficient Fine Tuning of Foundation Models.** Transformer-based foundation models [63, 15] have been developed as an improvement over traditional convolution-based architectures

[29, 31] for computer vision tasks. The transformer-based models achieve strong generalization performance [40] after training on large datasets. Moreover, the pre-trained transformers can be fine-tuned in limited data settings leading to state-of-the-art performance [32, 27]. As a computation, memory, and parameter-efficient alternative to full fine-tuning of such large pre-trained transformers, different Parameter Efficient Fine Tuning (PEFT) approaches have been developed. PEFT techniques freeze most of the large transformer backbone, fine-tune the remaining backbone parameters and/or introduce lightweight extra modules for adapting to the downstream task. Existing approaches can broadly be categorized as extra-module-based [72, 54], partial-tuning-based [71, 9], and visual prompt tuning-based [68, 68, 28] methods. Extra-module-based methods (*e.g.,* Adapter [51], side-tune[72]) introduce small additional learnable modules and keep the pre-rained backbone frozen. Partial-tuning-based methods (*e.g.,* Bias [9]) keep a large portion of the backbone frozen, and fine-tune only part of the foundation model to downstream tasks. Visual prompt-based PEFT methods (VPT) introduce a learnable prompt variable along with a learnable classification head over the fixed pre-trained backbone to be adapted to the downstream task. VPT [32] has shown significant improvement over other PEFT techniques, and can even outperform full fine-tuning in multiple datasets/settings [28].

**Calibration in Deep Learning Models**   Calibration methods have been increasingly explored to achieve trustworthy deep-learning models. Post-hoc calibration methods [26, 43, 73] aim to learn a calibration map for a standard trained deep learning model such that the map can transform the poorly calibrated probabilities to calibrated probabilities. Regularization-based calibration approaches introduce explicit regularization (such as with $L_2$ regularization [26], entropy regularization [50]), or implicit regularization (such as with focal loss [39]) during training to ensure that the trained model is calibrated. Data augmentation methods such as Label smoothing [44], and mixup training [60][74] have also been explored for developing calibrated deep learning models. Recent survey [65] provides a discussion of the most relevant works towards developing calibrated deep learning models. Most existing calibration methods are designed to tackle the over-confidence issue, which is more commonly observed for large models trained from limited data due to overfitting. We observe that fine-tuned foundation models exhibit severe under-confidence in their predictions, where existing calibration techniques are less effective. To this end, we propose a lightweight Bayesian framework that fills this critical gap.

**Few-Shot Adaptation and Relationship with Meta-Learning.**   In this work, we consider few-shot adaption with a focus on $N$-way $K$-shot classification [64, 22], where the model is presented with a few-shot training set with $N$-class, each having $K$ examples. For instance, 1-shot Cifar100 training set consists of 1 sample from each of the 100 classes. The model is then evaluated on the test set, which is identical to the query set in the meta-testing tasks [58]. It is worth to note that meta-learning (*e.g.,* matching networks [64], MAML [16], VERSA [23], PLATIPUS [17]) leverages an episodic learning paradigm to achieve few-shot adaptation, where both meta-training and meta-testing are done on the task level in an episodic fashion [69, 37] with a large number of $N$-way $K$-shot training tasks. In this work, we consider more challenging few-shot adaptation tasks (*e.g.,* 100-way 1-shot in Cifar100 and 102-way 1-shot in Flowers102) compared to the commonly used 5-way 1-shot meta-learning tasks. We leverage the power of the pre-trained foundation models, which eliminates the need of task based episodic meta-training. From a meta-learning perspective, the pre-training phase for the foundation model could be viewed as performing meta-knowledge acquisition similar to meta-training. The pre-trained model can be seen as an expert equipped with the meta-knowledge, and parameter-efficient fine-tuning performs quick adaptation to the downstream tasks, analogous to the support-set based adaptation done in meta-testing.

## 3   Bayesian Parameter-Efficient Fine-Tuning of Foundation Models

We start by introducing some fundamental concepts from evidential learning, which will serve as key building blocks in the proposed `Bayesian-PEFT` framework. We then detail the two Bayesian components: base rate adjustment to address under-confidence and diversity-inducing evidential ensemble to improve the reliability on both prediction accuracy and uncertainty quantification.

### 3.1   Preliminaries

Evidential Deep Learning (EDL) [56] introduces a computationally efficient framework to transform deterministic deep learning (DL) models into uncertainty-aware models. The key idea is to introduce a higher-order conjugate prior distribution over the predicted likelihood distribution and train the DL model to output parameters of the higher-order distribution. Towards classification, EDL models

[56, 11] introduce Dirichlet prior distribution for the multinomial likelihood distribution. Specifically, the output softmax layer of the DL model is replaced by a monotonic, non-negative transformation function (*e.g.,* ReLU, SoftPlus, or $\exp$) to obtain the evidence for different classes that are transformed into the Dirichlet parameters. Mathematically, for a DL model $f_\theta(\cdot)$, and an input sample $\mathbf{x}$, we have

$$e_i = \mathcal{E}\big(f_\theta(\mathbf{x})\big)_i \quad \alpha_i = e_i + a_i \times W \tag{1}$$

where $e_i$ is the output evidence for the $i^{\text{th}}$ class from the model $f_\theta(\cdot)$ and input sample $\mathbf{x}$, $a_i$ is the base rate for the $i^{\text{th}}$ class, $W$ is the non-informative prior weight usually set to the number of classes, $\mathcal{E}$ is the non-negative transformation function, and $\alpha_i$ parameterizes a Dirichlet distribution. Existing EDL works usually adopt a non-informative base rate of $a_i = \frac{1}{N} \forall i \in [1, N]$. Furthermore, a multinomial distribution $\texttt{Mult}(y|\mathbf{p})$ over labels is parameterized as $\mathbb{E}[p_i] = \frac{\alpha_i}{S}$, where the total Dirichlet Strength $S = \sum_{i=1}^{N} \alpha_i$.

An evidential model can be trained via a Type-II Maximum Likelihood-based evidential loss $\mathcal{L}^{\texttt{Log}}(\boldsymbol{x}, \boldsymbol{y})$ [56] with KL regularization that penalizes evidence assigned to non-ground-truth classes:

$$\mathcal{L}_{\texttt{evid}}(\boldsymbol{x}, \boldsymbol{y}) = \mathcal{L}^{\texttt{Log}}(\boldsymbol{x}, \boldsymbol{y}) + \lambda \text{KL}\big(\texttt{Dir}(\boldsymbol{p}|\tilde{\boldsymbol{\alpha}}) \| \texttt{Dir}(\boldsymbol{p}|\mathbf{1})\big) \tag{2}$$

where $\tilde{\boldsymbol{\alpha}} = \boldsymbol{y} + (\mathbf{1} - \boldsymbol{y}) \odot \boldsymbol{\alpha}$. Once trained, the evidential model can predict an evidence vector $\boldsymbol{e} = (e_1, e_2, ...e_N)^\top$ for a given test sample $\boldsymbol{x}$. From the predicted evidence, we obtain the model's belief ($\mathbf{b}$) over different classes, the correct belief $b_{\texttt{cor}}$, incorrect belief $b_{\texttt{inc}}$, and vacuity $u$ as

$$\mathbf{b} = \frac{\mathbf{e}}{S} \quad , \quad b_{\texttt{cor}} = \sum \mathbf{y} \odot \mathbf{b} \quad , \quad b_{\texttt{inc}} = \sum (\mathbf{1} - \mathbf{y}) \odot \mathbf{b} \quad , \quad u = \frac{N}{S}, \tag{3}$$

where vacuity $u$ is a second-order uncertainty [33] that captures the model's lack of knowledge in its prediction; $\mathbf{b}_{\texttt{cor}}$ and $\mathbf{b}_{\texttt{inc}}$ quantify model accuracy and error, respectively. However, neither $\mathbf{b}_{\texttt{cor}}$ nor $\mathbf{b}_{\texttt{inc}}$ can be evaluated without the ground-truth label, which is not available in the testing phase. Existing theoretical work has established an important connection between $\mathbf{b}_{\texttt{inc}}$ and dissonance [46], which is another second-order uncertainty [33] that can be quantified without the ground-truth label. More specifically, dissonance $\texttt{dis}$ can be evaluated as

$$\texttt{dis} = \sum_{n=1}^{N} \left( b_n \frac{\sum_{j \neq n} b_j Bal(b_j, b_n)}{\sum_{j \neq n} b_j} \right), \quad Bal(b_j, b_n) = \begin{cases} 2^{\frac{\min(b_j, b_n)}{b_j + b_n}}, & \text{if } b_i b_j > 0 \\ 0, & \text{otherwise} \end{cases} \tag{4}$$

where $Bal(\cdot, \cdot)$ is the relative mass balance function between two belief masses. The dissonance essentially captures the conflicting belief assigned to different classes [57].

### 3.2 Strengthening the prior belief through base rate adjustment

To gain a deeper understanding of the under-confident few-shot adaptation behavior of foundation models, we perform fine-grained uncertainty analysis using the predicted evidence from an evidential model. To this end, we replace the softmax layer in a VPT fine-tuned transformer model with an exponential-based evidential head that outputs non-negative evidence for different classes. We analyze the output evidence from the evidential model that reveals some interesting insights.

**Why is the model accurate?** First, we observe that the relative order of the evidence assigned to different classes is accurate. This implies that the model outputs relatively greater evidence for the correct class compared to all other classes that ensure the model's strong predictive performance. To more precisely quantify the model's accuracy, we adapt the lower bound of incorrect belief theorem developed for meta-learning [46] to evaluate the model accuracy through its predicted dissonance:

**Theorem 1.** *Consider an evidential model that outputs incorrect belief of $b_{inc}$ and the dissonance in the beliefs is $\texttt{dis}$. Then, the incorrect belief of the model will be at least half of the dissonance for all predictions from the evidential model.*

$$\frac{1}{2}\texttt{dis} \leq b_{inc} \quad where \quad 0 \leq \texttt{dis} \leq 1 \quad \& \quad 0 \leq b_{inc} \leq 1 \tag{5}$$

Figure 3a shows the test accuracy vs. dissonance curve, which is aligned with the relationship

between the incorrect belief and the dissonance given in the theorem above, where a low dissonance implies a low $b_{\texttt{inc}}$ (or a high accuracy). From all the testing samples, we observe relatively low dissonance and the highest is only slightly above 0.7 as shown in the figure. We further evaluate the Area Under the Curve of the Accuracy vs. $(1 - \texttt{dis})$ and obtain an AUC of 0.82 as shown in Figure 3b. This implies the model is able to clearly discriminate the ground-truth label from the rest without much confusion (*i.e.,* low dissonance) which ensures its good prediction accuracy.

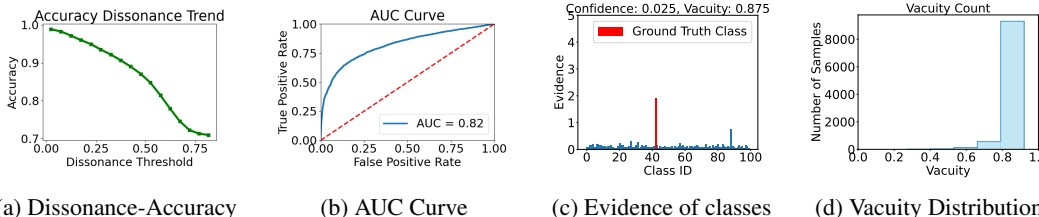

Figure 3: 1-shot Cifar10 results and evidence vacuity trends

**Why is the model under-confident?**    Despite being able to assign relatively more evidence to the correct label over the rest, it is also interesting to observe that the model generally assigns very low evidence to all the labels, including the correct one. Figure 3c shows the evidence distribution of one representative test data sample from Cifar100. As can be seen, most classes are assigned very low evidence that is close to zero. The ground-truth class is assigned higher evidence, but it is far from sufficient to make the prediction confident. The resultant confidence is only 0.025 while the vacuity is extremely high at 0.875, implying that the model *believes* it has very limited knowledge of the data sample despite it correctly identifying the correct label. Figure 3d shows the predicted vacuity over all the test samples, most of which are assigned a very high vacuity. This confirms the overly conservative behavior of the model, where *the low confidence is primarily due to the insufficient allocation of evidence* in its predictions. On the other hand, since the model is fairly accurate, it is reasonable to believe that the model underestimates the contribution of the rich prior knowledge gained through pre-training.

**Base rate adjustment to strength the prior belief.**    The fine-grained uncertainty analysis through the lenses of evidential learning not only explains the good predictive performance of the few-shot adapted model through PEFT but also unveils the root cause for its under-confident behavior, which is under-estimation of the contribution from the prior knowledge to the downstream task. While the classical Bayes' theorem offers a principal idea to address the issue, which is to strengthen the prior belief, there is a lack of practical way to achieve this. To this end, we propose to leverage the base rate introduced by the subjective logic theory [33] as an effective vehicle to adjust the prior belief gained through pre-training. According to (1), adjusting the base rate has the effect of changing the Dirichlet parameter $\alpha$, which will change the confidence for the prediction given by $\max_i \mathbb{E}[p_i]$. However, base rate adjustment needs to meet two key requirements: (1) the relative order of the Dirichlet parameters assigned to different classes should be preserved so that the predictive performance of the model remains unaffected, (2) the gap between the Dirichlet parameters for different classes is transformed such that the model becomes more confident in its predictions, making it well-calibrated. To meet these requirements, we we propose a transformation function $\mathcal{A}_m$ to the model's output evidence such that the model is well calibrated without any compromise in the generalization performance:

$$\boldsymbol{\alpha} = \mathcal{A}_m\big(f_\theta(\mathbf{x_i})\big) = \mathbf{e} + W\boldsymbol{\chi} \quad , \quad \chi_i = a_i^{\texttt{adj}} = \left(\frac{e_i - e_{\texttt{min}}}{e_{\texttt{min}}}\right)^m \tag{6}$$

where $\boldsymbol{\chi} = (\chi_1, \chi_2, ...\chi_N)^\top$ is the adjusted base rate, and $m \geq 1$ controls the base rate transformation. The adjusted base rate $\boldsymbol{\chi}$ considers evidence of all classes as a reference via $e_{\texttt{min}}$, and transforms the gap between different class evidences such that the model is well calibrated.

**Lemma 2.** *The base-rate adjusted model that uses learnable base rate $\boldsymbol{\chi} = (\chi_1, \chi_2, ..., \chi_N)^\top$ has the same generalization performance compared to the model using fixed base rate of $a_i = \frac{1}{N} \forall i \in [1, N]$*

**Theorem 3.** *For any $m \geq 1$, the transformation function $\mathcal{A}_m$ transforms the base rate for the class with the highest evidence $e_{\texttt{max}}$ and class with the second highest evidence $e_{\texttt{2nd}}$ such that the gap in Dirichlet parameters between the two classes is non-decreasing.*

**Remark.**  Theorem 3 ensures that the expected probability $\mathbb{E}[p_i]$ for the predicted class $i$ has an increased gap with the rest of the classes, which results in an increase of the model's confidence. Therefore, if the prediction is accurate, the model's calibration performance will be improved. Meanwhile, Lemma 2 ensures that the good prediction accuracy of the model is maintained by the proposed base rate adjustment strategy. The detailed proofs are given in Appendix D.

### 3.3   Building A Diversity Induced Evidential Ensemble

The second Bayesian component of the proposed B–PEFT framework aims to further improve the reliability of both prediction accuracy and uncertainty quantification when performing few-shot adap-

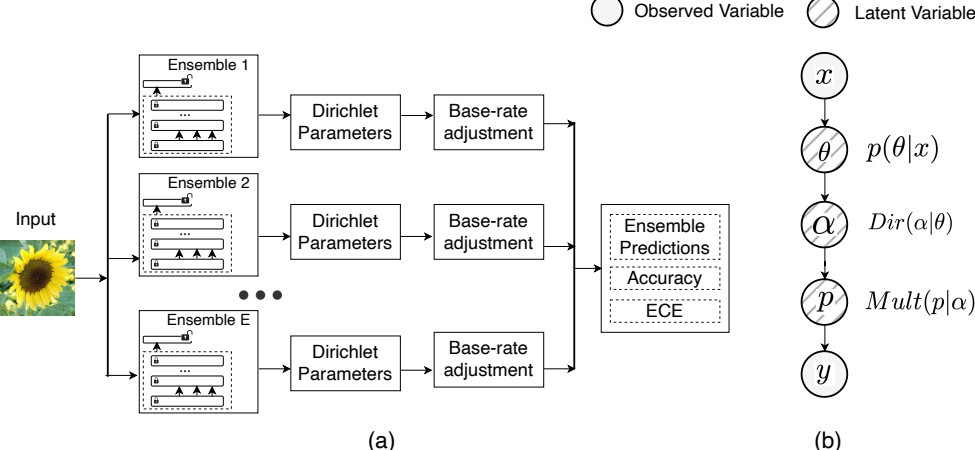

(a)                                    (b)

Figure 4: (a) Schematic diagram and (b) Graphical model of the B–PEFT model

tation. It performs Bayesian model averaging by building a diversity-inducing evidential ensemble. The ensemble of deep learning models (*i.e.,* deep ensemble) [20, 36] can effectively improve the generalization performance of deep learning models. Moreover, deep ensembles can capture the model uncertainty [36, 53] via the agreement-disagreement between the ensemble components. Model uncertainty essentially captures the uncertainty in the model parameters, which is denoted as $\theta$ of the graphical model of B–PEFT as shown in Figure 4(b). The schematic diagram of B–PEFT is shown in Figure 4(a). The model uncertainty can be leveraged to evaluate the reliability of fine-grained uncertainty output by the evidential model.

The effectiveness of the ensembles has been empirically demonstrated across multiple datasets/settings [30] with theoretical guarantees [2]. However, standard deep ensembling leads to limited diversity among the ensemble components as it only considers the random initialization of components. We propose a novel diversity-inducing ensembling scheme for the evidential models. Similar to the deep ensemble [36], we also consider randomly initialized evidential models. We additionally train each ensemble component with different strengths for incorrect evidence regularization along with evidential loss objective. The overall objective for each ensemble component is identical to (2).

However, each ensemble component is trained with different incorrect evidence (or belief) regularization strengths (*i.e.,* different components place different priorities for the minimization of incorrect evidence over the maximization of correct evidence) which leads to diversity among the components. Since each component's priority for minimizing the incorrect evidence is different, the components focus on different attributes/features in the data that help the model avoid overfitting to an identical set of discriminative features. As a result, the proposed evidential ensembling scheme implicitly pushes the ensemble components away from each other, making it equivalent to the repulsive force in the Stein Variational Gradient Descent (SVGD) [12, 13].

**Lemma 4.** *For given incorrect evidence regularization $\mathcal{L}_{reg}^{inc}$, and E ensemble components with regularization strengths $\lambda_p, p \in [1, P]$, the ensemble components in the evidence space are implicitly pushed away from each other by a force $\lambda_p \nabla \mathcal{L}_{reg}^{inc}$ that acts identical to the repulsive force in Stein Variational Gradient Descent (SVGD) based ensembles.*

**Remark.** The detailed proof is given in the Appendix. We present an intuitive visualization of the update of the evidential ensemble model for different strengths ($\lambda_1 < \lambda_2 < ... < \lambda_P$) of incorrect evidence regularization for different seeds in Figure 5. Each ensemble component aims to maximize the likelihood (direction $\vec{A}$) and minimize incorrect

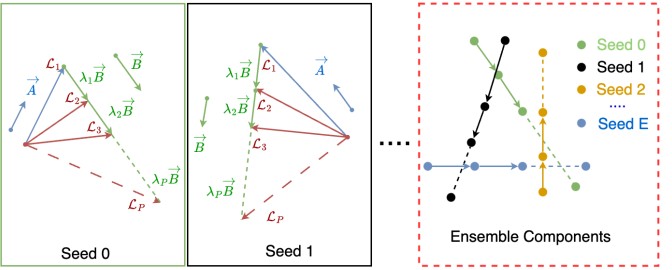

Figure 5: Illustration of ensemble diversity achieved through incorrect belief regularization with different strength

Table 1: Prediction accuracy and ECE performance on few-shot adaptation

| K (Shot) | Cifar10 | | Cifar100 | | Food101 | | Flowers102 | |
|---|---|---|---|---|---|---|---|---|
| | Accuracy ↑ | ECE ↓ | Accuracy ↑ | ECE ↓ | Accuracy ↑ | ECE ↓ | Accuracy ↑ | ECE ↓ |
| **(a) Standard Model** | | | | | | | | |
| 1-Shot | $69.578_{\pm1.351}$ | $0.437_{\pm0.010}$ | $48.637_{\pm0.757}$ | $0.393_{\pm0.008}$ | $35.702_{\pm1.095}$ | $0.263_{\pm0.009}$ | $88.161_{\pm0.91}$ | $0.61_{\pm0.004}$ |
| 2-Shot | $81.771_{\pm1.333}$ | $0.400_{\pm0.016}$ | $64.501_{\pm0.303}$ | $0.494_{\pm0.002}$ | $53.954_{\pm0.659}$ | $0.39_{\pm0.004}$ | $93.462_{\pm1.072}$ | $0.55_{\pm0.006}$ |
| 5-Shot | $88.707_{\pm0.423}$ | $0.255_{\pm0.008}$ | $76.758_{\pm0.525}$ | $0.517_{\pm0.001}$ | $65.586_{\pm0.197}$ | $0.424_{\pm0.002}$ | $97.363_{\pm0.165}$ | $0.472_{\pm0.013}$ |
| 10-Shot | $91.061_{\pm0.217}$ | $0.212_{\pm0.005}$ | $80.720_{\pm0.329}$ | $0.501_{\pm0.003}$ | $71.566_{\pm0.069}$ | $0.444_{\pm0.003}$ | $98.244_{\pm0.114}$ | $0.439_{\pm0.018}$ |
| 20-Shot | $92.678_{\pm0.37}$ | $0.166_{\pm0.004}$ | $82.608_{\pm0.266}$ | $0.487_{\pm0.004}$ | $74.914_{\pm0.178}$ | $0.460_{\pm0.003}$ | $98.431_{\pm0.100}$ | $0.425_{\pm0.017}$ |
| **(b) Evidential Model** | | | | | | | | |
| 1-Shot | $70.197_{\pm1.013}$ | $0.557_{\pm0.011}$ | $51.127_{\pm0.435}$ | $0.499_{\pm0.004}$ | $36.297_{\pm1.407}$ | $0.349_{\pm0.014}$ | $89.225_{\pm1.03}$ | $0.846_{\pm0.004}$ |
| 2-Shot | $81.613_{\pm1.736}$ | $0.553_{\pm0.01}$ | $65.545_{\pm0.339}$ | $0.620_{\pm0.004}$ | $52.855_{\pm0.551}$ | $0.485_{\pm0.005}$ | $95.071_{\pm0.413}$ | $0.874_{\pm0.006}$ |
| 5-Shot | $88.764_{\pm0.896}$ | $0.391_{\pm0.015}$ | $77.561_{\pm0.716}$ | $0.744_{\pm0.006}$ | $65.135_{\pm0.27}$ | $0.536_{\pm0.005}$ | $97.602_{\pm0.199}$ | $0.686_{\pm0.02}$ |
| 10-Shot | $92.014_{\pm0.353}$ | $0.388_{\pm0.006}$ | $81.561_{\pm0.291}$ | $0.765_{\pm0.002}$ | $70.863_{\pm0.261}$ | $0.673_{\pm0.003}$ | $98.326_{\pm0.233}$ | $0.444_{\pm0.008}$ |
| 20-Shot | $93.029_{\pm0.239}$ | $0.360_{\pm0.015}$ | $83.100_{\pm0.184}$ | $0.782_{\pm0.001}$ | $72.060_{\pm0.309}$ | $0.599_{\pm0.003}$ | $98.708_{\pm0.014}$ | $0.411_{\pm0.013}$ |
| **(c) Base-rate adjusted Evidential Model (Calibrated Evidential Model)** | | | | | | | | |
| 1-Shot | $70.197_{\pm1.013}$ | $0.027_{\pm0.002}$ | $51.127_{\pm0.435}$ | $0.077_{\pm0.004}$ | $36.297_{\pm1.407}$ | $0.081_{\pm0.011}$ | $89.225_{\pm1.03}$ | $0.025_{\pm0.004}$ |
| 2-Shot | $81.613_{\pm1.736}$ | $0.040_{\pm0.013}$ | $65.545_{\pm0.339}$ | $0.08_{\pm0.003}$ | $52.855_{\pm0.551}$ | $0.063_{\pm0.006}$ | $95.071_{\pm0.413}$ | $0.023_{\pm0.003}$ |
| 5-Shot | $88.764_{\pm0.896}$ | $0.028_{\pm0.006}$ | $77.561_{\pm0.716}$ | $0.044_{\pm0.002}$ | $65.135_{\pm0.270}$ | $0.037_{\pm0.003}$ | $97.602_{\pm0.199}$ | $0.015_{\pm0.002}$ |
| 10-Shot | $92.014_{\pm0.353}$ | $0.019_{\pm0.001}$ | $81.561_{\pm0.291}$ | $0.034_{\pm0.002}$ | $70.863_{\pm0.261}$ | $0.054_{\pm0.002}$ | $98.326_{\pm0.233}$ | $0.023_{\pm0.003}$ |
| 20-Shot | $93.029_{\pm0.239}$ | $0.016_{\pm0.002}$ | $83.100_{\pm0.184}$ | $0.031_{\pm0.001}$ | $72.060_{\pm0.309}$ | $0.050_{\pm0.002}$ | $98.708_{\pm0.014}$ | $0.021_{\pm0.000}$ |
| **(d) B-PEFT Model (Ours)** | | | | | | | | |
| 1-Shot | $\mathbf{74.674_{\pm0.968}}$ | $\mathbf{0.024_{\pm0.002}}$ | $\mathbf{52.335_{\pm0.610}}$ | $\mathbf{0.067_{\pm0.001}}$ | $\mathbf{38.745_{\pm0.184}}$ | $\mathbf{0.021_{\pm0.001}}$ | $\mathbf{90.238_{\pm0.101}}$ | $\mathbf{0.023_{\pm0.001}}$ |
| 2-Shot | $\mathbf{83.865_{\pm0.735}}$ | $\mathbf{0.022_{\pm0.002}}$ | $\mathbf{67.563_{\pm0.272}}$ | $\mathbf{0.056_{\pm0.001}}$ | $\mathbf{54.661_{\pm0.017}}$ | $\mathbf{0.020_{\pm0.001}}$ | $\mathbf{95.715_{\pm0.020}}$ | $\mathbf{0.021_{\pm0.002}}$ |
| 5-Shot | $\mathbf{90.556_{\pm0.160}}$ | $\mathbf{0.017_{\pm0.001}}$ | $\mathbf{80.081_{\pm0.067}}$ | $\mathbf{0.036_{\pm0.000}}$ | $\mathbf{66.548_{\pm0.110}}$ | $\mathbf{0.034_{\pm0.001}}$ | $\mathbf{97.807_{\pm0.066}}$ | $\mathbf{0.014_{\pm0.002}}$ |
| 10-Shot | $\mathbf{92.956_{\pm0.086}}$ | $\mathbf{0.014_{\pm0.000}}$ | $\mathbf{83.038_{\pm0.045}}$ | $\mathbf{0.031_{\pm0.000}}$ | $\mathbf{71.661_{\pm0.212}}$ | $\mathbf{0.038_{\pm0.002}}$ | $98.050_{\pm0.041}$ | $\mathbf{0.011_{\pm0.001}}$ |
| 20-Shot | $\mathbf{93.833_{\pm0.021}}$ | $\mathbf{0.014_{\pm0.001}}$ | $\mathbf{83.748_{\pm0.065}}$ | $\mathbf{0.030_{\pm0.001}}$ | $\mathbf{75.495_{\pm0.128}}$ | $0.043_{\pm0.001}$ | $98.193_{\pm0.020}$ | $\mathbf{0.010_{\pm0.001}}$ |

evidence (direction $\vec{B}$). The strengths of incorrect evidence regularization (direction $\vec{B}$) are different for each ensemble component that acts as an implicit repulsive force among the ensemble components, ensuring that they are diverse from each other. Different from the SVGD-based ensemble, in our proposed model, the particles do not need to explicitly communicate with each other making our proposed approach computationally efficient, scalable, and generalizable.

## 4    Experiments and Results

**Experiment setup, datasets, and baselines.**    We consider $K$-shot adaptation (*i.e.,* the dataset has $K$ examples per class in the training set) with Cifar10 [1], Cifar100 [1], Food101 [8], and Flowers102 [45] datasets. For instance, the 2-shot Cifar100 dataset has 2 examples per class leading to a total of 200 labeled training samples. For all datasets and experiments, the training set is a few-shot dataset, and the evaluation is done on the standard test set available with benchmark datasets. Details of the few-shot training datasets along with additional experiment details are presented in the Appendix E. We consider large pre-trained vision transformer with ViT backbone [15] and consider Visual Prompt Tuning (VPT) [32], along with bias fine-tuning [9] and adapter fine-tuning [72] as the PEFT techniques (We use VPT as the representative PEFT where not specified due to its superior performance). We consider accuracy-preserving post-hoc calibration techniques including Temperature Scaling (TS) [26], Parameterized Temperature Scaling (PTS) [61], and Isotonic Regression (IR-MC) [6] as the baseline calibration techniques.

**Prediction and calibration performance.**    We first consider standard cross-entropy (CE)-based PEFT of the supervised pre-trained ViT model on few-shot datasets. We present the accuracy and calibration results of VPT in Table 1 (a). We observe that the straightforward adaption of the models leads to accurate but under-confident models as indicated by a high ECE. The evidential models as shown in Table 1 (b) have comparable or better generalization performance across the datasets/settings. However, these models are also severely under-confident similar to CE-based models indicated by high ECE and accuracy-confidence trends (see Figure 11a, 11b in the Appendix). The overall performance of the calibrated evidential model using base-rate adjustment is presented in Table 1 (c). As can be seen, the accuracy remains the same as the adjusted base rate expands the gap between evidence of the class and preserves the relative order in the predicted class evidence. It effectively tackles the under-confidence issue, which leads to a significant improvement in the overall ECE performance across the datasets and settings (also see Figure 11c in the Appendix). Table 1 (d) shows

the results of the proposed B–PEFT model that introduces a diversity-enforcing ensemble of calibrated evidential models trained with different strengths of incorrect evidence regularization. performance.

Such a learning signal helps the model avoid overfitting (addressing the potential overconfidence issue) and leads to further improvement in generalization and the calibration.

We further carry out experiments with additional PEFT techniques of bias and adapter fine-tuning and on the $K$-shot Cifar100 dataset in Table 2. We observe that these PEFT techniques lead to a lower generalization performance (as indicated by a lower test accuracy) compared to the VPT-based technique. Nevertheless, the same underconfidence issue remains as shown in Table 2 (a) for standard cross-entropy trained model performance, and in Table 2 (b) for their evidential extensions. We then carry out experiments using base-rate adjusted evidential model and our proposed B–PEFT model. The results are shown in Table 2 (c) and Table 2 (d). Base-rate adjusted evidential model and B–PEFT are equally effective with these PEFT techniques, and they address the underconfidence issue, which leads to improvement in both generalization and calibration.

Table 2: Adapter and bias fine tuning results

| K (Shot) | Bias | | Adapter | |
|---|---|---|---|---|
| | Accuracy ↑ | ECE↓ | Accuracy ↑ | ECE ↓ |
| **(a) Standard Model** | | | | |
| 1-Shot | $35.514_{\pm 2.420}$ | $0.296_{\pm 0.023}$ | $46.150_{\pm 1.150}$ | $0.386_{\pm 0.010}$ |
| 2-Shot | $55.098_{\pm 4.932}$ | $0.384_{\pm 0.036}$ | $66.789_{\pm 0.514}$ | $0.513_{\pm 0.003}$ |
| 5-Shot | $74.203_{\pm 0.467}$ | $0.383_{\pm 0.002}$ | $78.738_{\pm 0.032}$ | $0.503_{\pm 0.000}$ |
| 10-Shot | $79.141_{\pm 0.233}$ | $0.336_{\pm 0.002}$ | $81.589_{\pm 0.031}$ | $0.470_{\pm 0.000}$ |
| **(b) Evidential Model** | | | | |
| 1-Shot | $36.243_{\pm 4.113}$ | $0.3498_{\pm 0.041}$ | $47.391_{\pm 1.421}$ | $0.463_{\pm 0.014}$ |
| 2-Shot | $58.258_{\pm 3.884}$ | $0.516_{\pm 0.032}$ | $67.523_{\pm 0.674}$ | $0.654_{\pm 0.006}$ |
| 5-Shot | $75.643_{\pm 0.698}$ | $0.509_{\pm 0.006}$ | $79.875_{\pm 0.051}$ | $0.670_{\pm 0.001}$ |
| 10-Shot | $80.158_{\pm 0.284}$ | $0.454_{\pm 0.001}$ | $82.674_{\pm 0.044}$ | $0.731_{\pm 0.001}$ |
| **(c) Base-rate adjusted Evidential Model** | | | | |
| 1-Shot | $36.243_{\pm 4.113}$ | $0.061_{\pm 0.011}$ | $47.391_{\pm 1.421}$ | $0.081_{\pm 0.005}$ |
| 2-Shot | $58.258_{\pm 3.884}$ | $0.077_{\pm 0.004}$ | $67.523_{\pm 0.674}$ | $0.070_{\pm 0.001}$ |
| 5-Shot | $75.643_{\pm 0.698}$ | $0.069_{\pm 0.002}$ | $79.875_{\pm 0.051}$ | $0.057_{\pm 0.000}$ |
| 10-Shot | $80.158_{\pm 0.284}$ | $0.063_{\pm 0.001}$ | $82.674_{\pm 0.044}$ | $0.052_{\pm 0.001}$ |
| **(d) B–PEFT Model (Ours)** | | | | |
| 1-Shot | $\mathbf{37.825}_{\pm 0.344}$ | $\mathbf{0.050}_{\pm 0.002}$ | $\mathbf{48.732}_{\pm 0.225}$ | $\mathbf{0.076}_{\pm 0.002}$ |
| 2-Shot | $\mathbf{62.796}_{\pm 1.080}$ | $\mathbf{0.065}_{\pm 0.005}$ | $\mathbf{69.187}_{\pm 0.153}$ | $\mathbf{0.068}_{\pm 0.002}$ |
| 5-Shot | $\mathbf{77.181}_{\pm 0.195}$ | $\mathbf{0.062}_{\pm 0.001}$ | $\mathbf{79.918}_{\pm 0.010}$ | $\mathbf{0.051}_{\pm 0.001}$ |
| 10-Shot | $\mathbf{80.788}_{\pm 0.064}$ | $\mathbf{0.059}_{\pm 0.008}$ | $\mathbf{82.748}_{\pm 0.016}$ | $\mathbf{0.049}_{\pm 0.001}$ |

**Comparison with existing calibration techniques.** We compare our proposed evidential calibra tion technique with existing calibration techniques using a representative Cifar100 dataset on different few-shot classification settings. We consider Isotonic Regression (IR-MC)[6], Temperature Scaling (TS) [26], and Parameterized Temperature Scaling (PTS) [61] as baselines. Overall results are presented in Table 3. The baseline model and its evidential extension are poorly calibrated. All calibration

Table 3: ECE performance comparison

| Model | 1 **Shot** | 2 **Shot** | 5 **Shot** | 10 **Shot** |
|---|---|---|---|---|
| CE Model [32] | 0.393 | 0.494 | 0.517 | 0.501 |
| Evidential Model [56] | 0.499 | 0.620 | 0.744 | 0.765 |
| TS [26] | 0.092 | 0.074 | 0.043 | 0.036 |
| PTS [61] | 0.145 | 0.129 | 0.096 | 0.083 |
| IR-MC [6] | 0.091 | 0.104 | 0.103 | 0.085 |
| BR-Evid (Ours) | 0.077 | 0.080 | 0,044 | 0.034 |
| B–PEFT **(Ours)** | **0.067** | **0.056** | **0.036** | **0.031** |

techniques transform the logits to address the under-confidence issue of the model. Our proposed Base-rate adjusted Evidential model (BR-Evid) addresses the under-confidence issue in the evidential model leading to significant improvement in the ECE. B–PEFT model further improves on the BR-Evid model as it effectively addresses both the under-confidence issue (via the base rate adjustment) and overconfidence issue (via the Bayesian ensembling) that leads to superior calibration results as indicated by the lowest ECE in all few-shot settings.

**Uncertainty quantification results.** We investigate the uncertainty quantification performance of our proposed calibrated evidential ensemble model. The developed model can reflect the model uncertainty (*i.e.,* the model's confidence in its predictions) through the ensemble agreement/disagreement and the distributional uncertainty through the sharpness of evidential prior distribution (*i.e.,* the vacuity). For the analysis, we use Cifar10 as in-distribution (ID) dataset for different shots, and Cifar100 as OOD dataset. As shown in Figure 6, we plot vacuity and variance distribution for 1 and 5 shots. A single model outputs the vacuity that can be used to detect OOD samples. We can see ID samples are in the lower vacuity region and OOD samples are in the higher vacuity region. As the number of shots increases, the region becomes more separate. However, for 1 shot, there are some OOD samples in the lower vacuity region as well. Since we can not trust vacuity alone for uncertainty, we utilize the variance of ensemble components to quantify model uncertainty. As we see in Figure 6(c), the variance of OOD samples mostly lies in the high variance region. As we increase the number of shots, the variance shifts towards the lower region. High variance indicates that model-predicted vacuity can not be totally trusted. This behavior is qualitatively shown in Figure

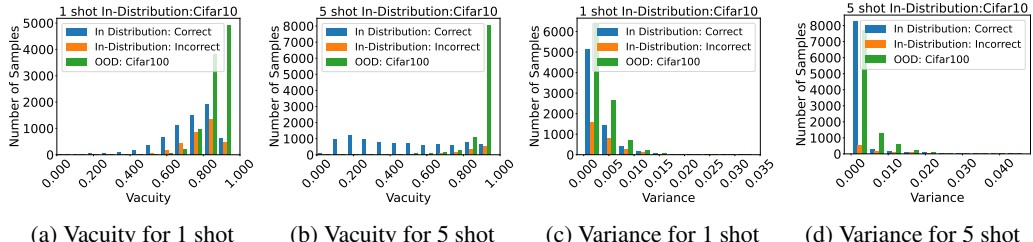

| (a) Vacuity for 1 shot | (b) Vacuity for 5 shot | (c) Variance for 1 shot | (d) Variance for 5 shot |

Figure 6: (a-b): Vacuity distribution of a single model and (c-d): variance distribution of ensemble models for 1/5 shots cifar10 as In-Distribution and Cifar100 as Out-of-Distribution dataset

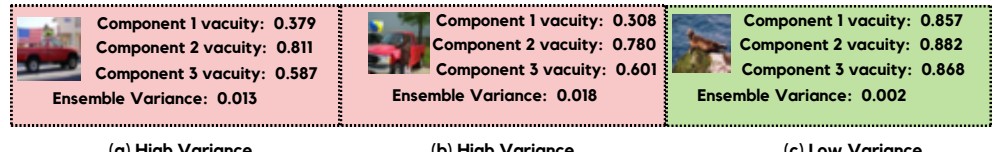

Figure 7: Qualitative analysis of OOD samples for 1-shot adaptation of Cifar10 as ID dataset and Cifar100 as OOD dataset

7. We observe that when only a single component outputs a low vacuity for the OOD samples in Figure 7 (a-b), the variance of the ensemble is high, implying a high model uncertainty. When all the components output a high vacuity to the OOD sample in Figure 7 (c), the variance is also low, implying a low model uncertainty.

**Ablation study.**   We carry out ablation with 1-shot Cifar100 dataset to study the impact of the base rate transformation order $m$ (Section 3.2) for different strengths of incorrect evidence regularization on the calibration performance. As we increase $m$, the probability gap between classes improves, leading to more confident predictions. However, the model starts to become overconfident for large $m$ values (see Figure 8).Moreover, with an increase in incorrect evidence regularization strength, the optimal $m$ value decreases to a smaller value (*e.g.,* optimal $m$ = 2.0 for $\lambda = 0.1$, and optimal $m = 1.5$ for $\lambda = 10$). Choosing a proper $m$ leads to the best-calibrated evidential model. We present additional results studying the impact of $m$ in Appendix F.1.

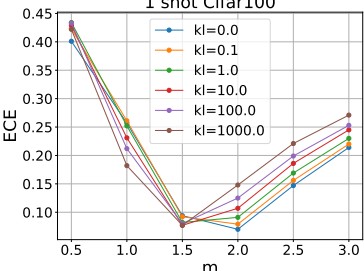

Figure 8: Impact of $m$

Limited by space, we provide results comparing meta-learning methods on standard few-shot tasks in Appendix F.4, discuss impact of various components (number of classes, data size, and unfrozen parameters) in Appendix F.5, and include additional experiments and comparisons, including OOD settings, in Appendix F.6. Moreover, we carry out additional ablation experiments to study the impact of incorrect evidence regularization strength (Appendix F.2), and the impact of ensemble components (Appendix F.3). We further carry out experiments and discuss applying PEFT to foundation models pre-trained in a self-supervised fashion (Appendix G). We discuss the societal impact and limitations of the work in Appendices H and I, respectively.

## 5   Conclusion

In this work, we focus on transformer-based large vision foundation models and investigate different parameter-efficient fine-tuning techniques for effective few-shot adaptation. We observe that the existing models are severely under-confident, especially in challenging datasets and settings. Moreover, existing models lack fine-grained uncertainty quantification capabilities. We extend the models to uncertainty-aware evidential models, and resort to the evidential framework to develop a novel Bayesian parameter efficient fine-tuning (B-PEFT) framework that integrates evidence-based base rate adjustment to addresses the under-confidence and a diversity inducing evidential ensemble technique to further improve the reliability in model prediction and uncertainty quantification. The B-PEFT framework possesses theoretically sound properties to ensure its superior generalization capability and robust calibration behavior. We carry out intensive experiments across different benchmark datasets and diverse few-shot settings that demonstrate the outstanding performance of B-PEFT.

## Acknowledgments

This research was supported in part by an NSF IIS award IIS-1814450. The views and conclusions contained in this paper are those of the authors and should not be interpreted as representing any funding agency. We would like to thank the anonymous reviewers for their constructive comments.

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

# Supplementary Material

# Appendix

## Table of Contents

# A  Organization of the Appendix

- In Section B, we present the table with summary of the key symbols used in this work.
- In Section C, we present related works in uncertainty-aware deep learning, describe evidential deep learning for classification in details, and discuss some key advantages of using evidential models to address the under-confidence issue over standard Bayesian models.
- In Section D, we present proof of all our theoretical claims.
- In Section E, we present the details of hyperparameters and additional experiment details.
- In Section F, we present additional experiment results including the impact of m for calibration performance, comparison results with meta-learning methods, the impact of the number of classes, data size, and unfrozen parameters, comparison in OOD settings, the impact of incorrect evidence regularization strength, the results of additional PEFT methods, and the impact of ensemble components.
- In Section G, we discuss the calibration and accuracy behavior for PEFT of large foundation models pre-trained in a self-supervised fashion.
- In Section H, we discuss societal impact of our work.
- In Section I, we discuss the limitations of our work and present potential future direction.

The source code for the experiments carried out in this work is attached in the supplementary materials and is available at the link: https://github.com/ritmininglab/B-PEFT

# B  Summary of the Symbols

Table 4: Summary of the symbols and their definitions

| Symbol | Definition |
|---|---|
| $\mathbf{x}$ | Input sample vector |
| $\mathbf{y}$ | Ground truth label as one hot vector |
| $\mathbf{e}, e_i$ | The evidence vector, and the evidence for class $i$ |
| $a_i$ | Fixed base rate for class $i$, usually set to $a_i = \frac{1}{N}$ |
| $\alpha_i$ | Dirichlet parameter value for class $i$ |
| $S = \sum_{i=1}^{N} \alpha_i$ | The Dirichlet Strength |
| $(\mathbf{b}, b_{\text{cor}}, b_{\text{inc}})$ | The belief vector, Correct belief, and the Incorrect belief |
| $N$ | Number of classes |
| $b_i$ | Belief for class i |
| $u$ | Vacuity output by the model |
| dis | Dissonance output by the model |
| $\mathcal{E}$ | Non-negative evidential transformation function (we use $\exp$) |
| $\chi_i$ | Learnable base rate for class $i$ |
| $\boldsymbol{\chi}$ | Learnable base rate vector |
| $\lambda$ | Incorrect evidence regularization strength |
| $\odot$ | Element wise multiplication between two vectors |
| $W$ | Non-informative prior weight |

# C  Further Discussion on Uncertainty-Aware Deep Learning

## C.1  Existing Uncertainty Quantification Methods in Deep Learning

Accurate quantification of predictive uncertainty is essential for the development of trustworthy Deep Learning (DL) models. To this end, DL models have been augmented to become uncertainty-aware using a variety of approaches such as ensemble-based approaches [36, 49], bayesian neural networks based approaches [42, 19, 7], and deterministic neural network based approaches [56, 11, 3]. Deep ensemble techniques [36, 49] construct an ensemble of neural networks, and the agreement/disagreement across the ensemble components is used to quantify different uncertainties. Alternatively, Bayesian neural networks [19][7][42] have been developed that consider a Bayesian formalism (*e.g.,* bayes-by-backprop [7], dropout during test [19]) to quantify different uncertainties.

ABNN [18] introduces Bayesian normalization layers after training of deep learning models, and requires additional training of these layers in a post-hoc manner. Deterministic neural network-based approaches [57, 41, 10] extend the existing neural network to become uncertainty-aware and enable the networks to quantify fine-grained uncertainties with a single forward pass through the network. Evidential deep learning models [56, 5, 75, 10, 10, 62], an instance of deterministic approaches, introduce a conjugate higher-order evidential prior for the likelihood distribution to enable the model to express the fine-grained uncertainties in both classification[56, 11] and regression problems [3, 47]. Towards classification, evidential models [56, 5, 75] introduce higher-order evidential Dirichlet prior to the multinomial likelihood that enables the deterministic neural network model to capture different uncertainty characteristics. In what follows, we first provide some additional details on using evidential learning model to perform classification. We then highlight some important advantage of using evidential models over standard Bayesian models in uncertainty quantification.

## C.2 Evidential Deep Learning Models for Classification

Evidential Deep Learning models, based on Subjective Logic theory [33], aim to train the model such that for any new input sample, the model can make predictions, as well as output fine-grained uncertainty information (via vacuity [56] and dissonance [34]). Towards capturing fine-grained uncertainty for classification problems, EDL models assume that the label for each sample is obtained from a generative process with a Dirichlet prior and a multinomial likelihood. The parameters for the Dirichlet prior express the vacuity and belief masses for uncertainty estimation. The conjugacy between the Dirichlet prior and the multinomial likelihood is explored, and different evidential losses are introduced for model training and inference [48]. In this work, we consider Type-II Maximum Likelihood-based evidential loss $\mathcal{L}^{\text{Log}}(\boldsymbol{x}, \boldsymbol{y})$ [56] with incorrect evidence regularization $\mathcal{L}_{\text{reg}}^{\text{inc}}(\boldsymbol{x}, \boldsymbol{y})$ given by [56]

$$\mathcal{L}_{\text{evid}}(\boldsymbol{x}, \boldsymbol{y}) = \mathcal{L}^{\text{Log}}(\boldsymbol{x}, \boldsymbol{y}) + \lambda \times \mathcal{L}_{\text{reg}}^{\text{inc}}(\boldsymbol{x}, \boldsymbol{y}) \tag{7}$$

We replace the softmax layer in the head of the VPT model with $\exp$ activation function. To avoid *zero evidence regions*, we also include the correct evidence regularization $\mathcal{L}_{\text{cor}}(\boldsymbol{x}, \boldsymbol{y}) = -\lambda_{\text{cor}} \log(\alpha_{gt} - 1)$ ( $\lambda_{\text{cor}}$ is the magnitude of the model's vacuity) in model training objective. The evidential model outputs evidence vector $\boldsymbol{e} = (e_1, e_2, ... e_N)$ for a given input $\boldsymbol{x}$ and corresponding ground truth label of $\boldsymbol{y}$. Based on the evidence, Dirichlet parameters are obtained as $\alpha_i = e_i + 1$. The Type-II Maximum likelihood-based evidential loss is given by

$$\mathcal{L}^{\text{Log}}(\boldsymbol{x}, \boldsymbol{y}) = -\ln \int \text{Mult}(\boldsymbol{y}|\boldsymbol{p})\text{Dir}(\boldsymbol{p}|\boldsymbol{\alpha})\mathrm{d}\boldsymbol{p} = \log S - \sum_{k=1}^{K} y_k \log \alpha_k \quad S = \sum_{k=1}^{K} \alpha_k \tag{8}$$

The incorrect evidence regularization guides the model to minimize the evidence for all classes other than the ground truth class and can take one of the following forms

1. KL-based incorrect evidence regularization term as in EDL [56]

$$\mathcal{L}_{\text{reg}}^{\text{EDL}}(\boldsymbol{x}, \boldsymbol{y}) = \text{KL}\big(\text{Dir}(\boldsymbol{p}|\tilde{\boldsymbol{\alpha}})||\text{Dir}(\boldsymbol{p}|\mathbf{1})\big)$$
$$= \log \Big( \frac{\Gamma \sum_{k=1}^{K} \tilde{\alpha}_k}{\Gamma(K) \prod_{k=1}^{K} \Gamma \tilde{\alpha}_k} \Big) + \sum_{k=1}^{K} (\tilde{\alpha}_k - 1)\Big[\psi(\tilde{\alpha}_k) - \psi\Big(\sum_{j=1}^{K} \tilde{\alpha}_j\Big)\Big] \tag{9}$$

   Where $\tilde{\boldsymbol{\alpha}} = \boldsymbol{y} + (\mathbf{1} - \boldsymbol{y}) \odot \boldsymbol{\alpha} = (\tilde{\alpha}_1, \tilde{\alpha}_2, ... \tilde{\alpha}_N)$ parameterize a dirichlet distribution, $\tilde{\alpha}_{i=gt} = 1, \tilde{\alpha}_i = \alpha_i \forall i \neq gt$, and $\odot$ represents element-wise product. Here, the KL regularization term encourages the Dirichlet distribution based on the incorrect evidence i.e., $\text{Dir}(\boldsymbol{p}|\tilde{\boldsymbol{\alpha}})$ to be flat which is possible when there is no incorrect evidence.

2. Incorrect evidence sum based regularization as in ADL [57]

$$\mathcal{L}_{\text{reg}}^{\text{ADL}}(\boldsymbol{x}, \boldsymbol{y}) = \sum_{k=1}^{K} \big(\mathbf{e} \odot (\mathbf{1} - \mathbf{y})\big)_k = \sum_{k=1}^{K} e_k \times (1 - y_k) \tag{10}$$

3. Incorrect belief-sum based regularization as in Units-ML [46]

$$\mathcal{L}_{\text{reg}}^{\text{Units}}(\boldsymbol{x}, \boldsymbol{y}) = \sum_{k=1}^{K} \big(\frac{\mathbf{e}}{S} \odot (\mathbf{1} - \mathbf{y})\big)_k = \sum_{k=1}^{K} \frac{e_k}{S} \times (1 - y_k) \tag{11}$$

All three regularizations guide the model to minimize the incorrect evidence(ideally close to zero). In our experiments, we consider KL-based incorrect evidence regularization.

## C.3 Evidential models vs. Standard Bayesian Models

As compared with the Bayesian-inspired models, evidential learning offers two key properties that allow us to formulate a principled solution to address the unique under-confident behavior of the PEFT methods. First, thanks to its evidence-based fine-grained uncertainty decomposition capability, we can separate two distinct sources of second-order uncertainty, including vacuity and dissonance. Different from the commonly used first-order uncertainty (e.g., entropy), these two second-order uncertainty serve as a key tool to understand why PEFT methods are both accurate (with a low dissonance) while being under-confident (with a high vacuity). This key insight suggests that these methods systematically under-estimate the contribution from the prior knowledge to the downstream task. While the classical Bayes' theorem offers a principal idea to address the issue, which is to strengthen the prior belief, there is a lack of practical way to achieve this. As the second key property, evidential learning allows us to leverage the base rate, which is rooted in the subjective logic theory as an effective vehicle to adjust the prior belief gained through pre-training. To this end, we propose a transformation function in Eq. (6) to adjust the base rate that leads to the increase of the model confidence while maintaining the predictive accuracy of the model as guaranteed by our theoretical results in Lemma 2 and Theorem 3. Furthermore, we develop belief-based diversity for ensemble of evidential models leading to the the B-PEFT model. In theory, evidential deep learning model could be augmented with the Bayesian normalization layers [18] or Bayesian neural networks [7] as an alternative to belief-based diversity of B-PEFT. We leave exploration of different techniques for diversity for Bayesian evidential model as a potential future work.

# D  Proofs of Theoretical Results

In this section, we provide the proofs of the major theoretical results presented in the main paper.

## D.1  Proof of Lemma 2

*Proof.* Consider an input sample $x$ for which the model outputs the evidence $(e_1, e_2, ..., e_N)^\top$. Let $e_{\max} = \max(e_1, e_2, ..., e_N)$, and $e_{\min} = \min(e_1, e_2, ..., e_N)$. Here, $e_{\max} \geq e_i \geq e_{\min} \forall i \in [1, N]$. For the evidential model with fixed base rate of $a_i = \frac{1}{N} \forall i \in [1, N]$, the model's predicted class is given by $c_{\mathtt{pred}} = \arg\max(e_1 + a_1 \times W, e_2 + a_2 \times W, ..., e_N + a_N \times W) = \arg\max(e_1 + 1, e_2 + 1, ..., e_N + 1) = \mathtt{Index}(e_{\max})$. For the calibrated model with learnable $\boldsymbol{\chi} = (\chi_1, \chi_2, ..., \chi_N)^\top$, the model's predicted class is given by $c_{\mathtt{pred}}^{\mathtt{new}} = \arg\max(\alpha_1, \alpha_2, ..., \alpha_N) = \arg\max(e_1 + \chi_1 \times W, e_2 + \chi_2 \times W, ..., e_N + \chi_N \times W) = \arg\max(e_1 + N(\frac{e_1}{e_{\min}}) - N, e_2 + N(\frac{e_2}{e_{\min}}) - N, ..., e_N + N(\frac{e_N}{e_{\min}}) - N)$. Since $e_{\max} \geq e_i \geq e_{\min} \forall i \in [1, N]$, $\alpha_{\max} \geq \alpha_i \geq \alpha_{\min} \forall i \in [1, N]$, and $c_{\mathtt{pred}}^{\mathtt{new}} = c_{\mathtt{pred}}$ $\qquad\square$

## D.2  Proof of Theorem 3

*Proof.* Consider an input sample $\mathbf{x}$ for which the model outputs the evidence $(e_1, e_2, ..., e_N)^\top$. Let $e_{\max} = \max(e_1, e_2, ...e_N)$, $e_{\min} = \min(e_1, e_2, ...e_N)$, and $e_{\max} \geq e_{\mathtt{2nd}} \geq, ..., \geq e_{\min}$. For the evidential model with a fixed base rate of $a_i = \frac{1}{N} \forall i \in [1, N]$, the difference between the Dirichlet parameters for class with maximum evidence and class with the second maximum evidence is given by $\alpha_{\max} - \alpha_{\mathtt{2nd}} = e_{\max} + a_{\max}W - e_{\mathtt{2nd}} - a_{\mathtt{2nd}}W = e_{\max} - e_{\mathtt{2nd}}$ as $a_i = \frac{1}{N} \forall i \in [1, N]$.

For the calibrated model with learnable $\boldsymbol{\chi} = (\chi_1, \chi_2, ..., \chi_N)$, the difference between the Dirichlet parameters for class with maximum evidence and class with the second maximum evidence is given by $\alpha_{\max} - \alpha_{\mathtt{2nd}} = e_{\max} + \chi_{\max}W - e_{\mathtt{2nd}} - \chi_{\mathtt{2nd}}W = (e_{\max} - e_{\mathtt{2nd}}) + (\chi_{\max} - \chi_{\mathtt{2nd}})W$. Now,

$$\chi_{\max} = \left(\frac{e_{\max} - e_{\min}}{e_{\min}}\right)^m = \left(\frac{e_{\max}}{e_{\min}} - 1\right)^m \quad \& \quad \chi_{\mathtt{2nd}} = \left(\frac{e_{\mathtt{2nd}} - e_{\min}}{e_{\min}}\right)^m = \left(\frac{e_{\mathtt{2nd}}}{e_{\min}} - 1\right)^m \tag{12}$$

$$\text{Or,} \left(\chi_{\max} - \chi_{\mathtt{2nd}}\right) = \left(\frac{e_{\max}}{e_{\min}} - 1\right)^m - \left(\frac{e_{\mathtt{2nd}}}{e_{\min}} - 1\right)^m \tag{13}$$

Since $\frac{e_i}{e_{\min}} \geq 1 \forall i \in [1, N]$, and $e_{\max} \geq e_{\mathtt{2nd}} \geq, ..., \geq e_{\min}$, $(\chi_{\max} - \chi_{\mathtt{2nd}}) \geq 0 \forall m > 0$. For $e_{\max} > e_{\mathtt{2nd}}$, $\& m > 0$, $\chi_{\max} - \chi_{\mathtt{2nd}} > 0$. Thus, with the proposed learnable base rate, the gap between

the two largest Dirichlet parameters is maintained whenever $m = 0$ and/or $e_{\max} = e_{\text{2nd}}$. Moreover, whenever $m \geq 1$ and $e_{\max} > e_{\text{2nd}}$, the Dirichlet parameter gap between the two classes is increased by a factor of $\left(\frac{e_{\max}}{e_{\min}} - 1\right)^m - \left(\frac{e_{\text{2nd}}}{e_{\min}} - 1\right)^m$. $\qquad\square$

### D.3 Connection with SVGD-based Bayesian Ensembling and Proof of Lemma 4

We first carry out an analysis of the update in Stein Variational Gradient Descent (SVGD) based ensembling [12, 13] that reveals the repulsive force acting among the ensemble components that pushes the particles away and introduces diversity. We then consider ensemble components with different strengths of incorrect evidence regularization $\mathcal{L}_{\text{reg}}^{\text{inc}}$ and analyze the update to the evidential model in the evidence space that reveals a repulsive diversity-enforcing force acting identical to the SVGD based ensemble.

SVGD update involves randomly initializing the particles and iteratively updating the particles to match the target distribution, which is summarized below.

---

**Algorithm 1** SVGD Update

**Input:** $\{x_i^0\}_{i=1}^N$: A set of initial parameters, and target distribution density function $p(x)$
**For $L$ iterations**, iteratively update the particles as

- $x_i^{l+1} = x_i^l + \epsilon_l \hat{\phi}^*(x_i^l)$, where $\hat{\phi}^*(x) = \frac{1}{E} \sum_{e=1}^E k(x_e^l, x) \nabla_{x_e^l} \log p(x_e^l) + \nabla_{x_e^l} k(x_e^l, x)$

Here, $\epsilon_l$ is the step size at iteration $l$, and $k(\cdot, \cdot)$ is the kernel function that measures similarity.
**Output:** $\{x_i^0\}_{i=1}^N$: A set of initial parameters, and target distribution density function $p(x)$

---

For given incorrect evidence regularization $\mathcal{L}_{\text{reg}}^{\text{inc}}$, and P ensemble components with regularization strengths $\lambda_p, p \in [1, P]$, the ensemble components in the evidence space are implicitly pushed away from each other by a force $\lambda_p \nabla \mathcal{L}_{\text{reg}}^{\text{inc}}$ that acts identical to the repulsive force in Stein Variational Gradient Descent (SVGD) based ensembles.

*Proof.* For simplicity, consider RBF kernel for $k(\cdot, \cdot)$ *i.e.,* $k(a, b) = \exp\left(\frac{-1}{h}(a - b)^2\right)$, two particles $x_1, x_2$, and analyze their updates in SVGD based ensembling. At iteration $l$, the update to particle $x_2$ is given by

$$\hat{\phi}^*(x_2^l) = \frac{1}{2}\Big(k(x_2^l, x_1^l)\nabla_{x_1^l} \log p(x_1^l) + k(x_2^l, x_2^l)\nabla_{x_2^l} \log p(x_2^l) + \nabla_{x_1^l} k(x_1^l, x_2^l) + \nabla_{x_2^l} k(x_1^l, x_2^l)\Big)$$

In the above update, $\overrightarrow{Q} = k(x_2^l, x_1^l)\nabla_{x_1^l} \log p(x_1^l) + k(x_2^l, x_2^l)\nabla_{x_2^l} \log p(x_2^l)$ aims to guide the particles in the direction that maximizes the likelihood, and the update direction $\overrightarrow{R} = \nabla_{x_1^l} k(x_1^l, x_2^l) + \nabla_{x_2^l} k(x_1^l, x_2^l)$ acts as the repulsive force. Considering the repulsive force

$$\overrightarrow{R} = \nabla_{x_1^l} k(x_1^l, x_2^l) + \nabla_{x_2^l} k(x_1^l, x_2^l) = \nabla_{x_1^l} \exp\left(\frac{-1}{h}(x_1^l - x_2^l)^2\right) + \nabla_{x_2^l} \exp\left(\frac{-1}{h}(x_1^l - x_2^l)^2\right)$$

$$= \frac{2}{h}(x_2^l - x_1^l)k(x_1^l, x_2^l)$$

As can be seen, the repulsive force $\overrightarrow{R}$ pushes the particle $x_2^l$ in the direction away from particle $x_1^l$ that introduces diversity. With more particles, each particle is updated in the direction that maximizes the likelihood, and the particle is pushed away from all other particles (by force $R$).

Next, consider ensemble components with different strengths of incorrect evidence regularization $\mathcal{L}_{\text{reg}}^{\text{inc}}$ to analyze the update to the evidential model in the evidence space. For simplicity, we consider the incorrect evidence sum-based regularization similar to ADL without correct evidence regularization (The analysis is valid for all incorrect evidence regularization and for models with correct evidence regularization). For a model with incorrect evidence regularization, the overall evidential loss is given by:

$$\mathcal{L}_{\text{evid}}(\boldsymbol{x}, \boldsymbol{y}) = \mathcal{L}^{\text{Log}}(\boldsymbol{x}, \boldsymbol{y}) + \lambda \times \mathcal{L}_{\text{reg}}^{\text{ADL}}(\boldsymbol{x}, \boldsymbol{y}) = \log S - \sum_{k=1}^K y_k \log \alpha_k + \lambda \times \sum_{k=1}^K e_k \times (1 - y_k)$$

The gradient of the loss with respect to logits (the output head layer, where $e_k = \exp(o_k)$) is given by

$$\text{grad}_k = \frac{\partial \mathcal{L}^{\text{Log}}(\mathbf{x}, \mathbf{y})}{\partial o_k} + \frac{\partial \mathcal{L}^{\text{ADL}}_{reg}(\mathbf{x}, \mathbf{y})}{\partial o_k} = \left(\frac{1}{S} - \frac{y_k}{\alpha_k}\right)\frac{\partial e_k}{\partial o_k} + \lambda \times (1 - y_k) \times \frac{\partial e_k}{\partial o_k} \quad (14)$$

$$= \left(\frac{1}{S} - \frac{y_k}{\alpha_k} + \lambda(1 - y_k)\right)e_k = \left(\frac{1}{S} - \frac{y_k}{\alpha_k} + \lambda(1 - y_k)\right)e_k \quad (15)$$

Consider $K$ class classification problem. The gradient update to the logit layer for the evidential model is given by

$$\text{grad}_k = e_k \times \begin{bmatrix} \frac{1}{S} - \frac{y_1}{\alpha_1} \\ \frac{1}{S} - \frac{y_2}{\alpha_2} \\ ... \\ \frac{1}{S} - \frac{y_K}{\alpha_K} \end{bmatrix} + e_k \times \lambda \times \begin{bmatrix} 1 - y_1 \\ 1 - y_2 \\ ... \\ 1 - y_K \end{bmatrix} \quad (16)$$

$$= \overrightarrow{A} + \lambda \times \overrightarrow{B} \quad (17)$$

Here, $y_k \in [0, 1], y_k = 1$ if $k = \texttt{gt}$, and $y_k = 0$ otherwise. Moreover, the $\lambda$ value is varied, and different $\lambda$ values lead to different evidential models.

The update force $\overrightarrow{A}$ pushes the evidential model in the direction that maximizes the likelihood (similar to $\overrightarrow{Q}$ in SVGD-based update), and the force $\overrightarrow{B}$ implicitly pushes the ensemble components away from each other (similar to the repulsive force $\overrightarrow{R}$ in SVGD-based update). Each component moves in $\overrightarrow{B}$ direction with a different force determined by the incorrect evidence regularization strength $\lambda$ that ensures that the ensemble components are diverse. Due to the different strengths of incorrect evidence regularization, each ensemble component places a different priority level for minimization of incorrect evidence over acquiring correct evidence, which ensures that the ensemble components remain diverse. □

## E   Dataset and Implementation Details

We consider ViT model backbone [15] that is pre-trained in a supervised fashion, and 4 benchmark datasets of Cifar10 [1], Cifar100 [1], Food101 [8], and Flowers102 [45]. We consider few-shot adaptation with $K$-shot classification problem (We experiment with $K$ values of 1, 2, 5, 10, and 20). The few-shot training set is constructed by randomly selecting $K$ samples per class from the training set of the benchmark datasets. We consider 2-shot validation set for all datasets and settings. We train the model on the few-shot training set, use the 2-shot validation set for hyperparameter tuning, and evaluate all models on the benchmark test set with all the test set samples. We augment the few-shot training set and the few-shot validation set with resize, random horizontal flip, and cropping. The dataset details are also presented in Table 5. We train all the models for 50 epochs on the few-shot training dataset with a batch size of 64 samples at each iteration and evaluate the model on the benchmark test set. The evidence is in the range [0, infinity], and some stability issues could potentially arise in extreme cases when the logit output is extremely low (i.e. close to negative infinity). In our experiments, we did not observe the stability issue. Still, the issue can arise in some extreme cases for which a small delta in the denominator could be introduced or the network's logits could be bounded to be greater than a small negative value. For the calibration baseline model of Parameterized Temperature Scaling (PTS) [61], we consider a 2-layer neural network with 128 nodes in the hidden layer (we carry out hyperparameter tuning with 1-layer, 2-layer, and 3-layer networks and select the best model), and train with a learning rate of 0.00001. For Temperature Scaling [26], we optimize for the temperature hyperparameter using Adam optimizer, and a learning rate of 0.01 (We also experiment with SGD optimizer, and learning rates of 0.1, 1.0, 0.01, and 0.0001 to select the best performing model). The evidential models use incorrect evidence regularization strength of (0, 0.1, 1.0, 10.0, 100.0, and 1000.0). For Isotonic Regression [6], we consider multi-class setting and sklearn package. We use VPT [32] as the representative PEFT where not specified due to its superior performance. The key model performance results are averaged across 5 different runs to present the mean and the standard deviation. The experiments use Pytorch and are carried out on a workstation with NVIDIA RTX A6000 GPU.

| Dataset Name | Number of Classes | Training Samples | Validation Samples | Test Samples |
|---|---|---|---|---|
| Cifar10 [1] | 10 | $10 \times K$ | 20 | $10,000$ |
| Cifar100 [1] | 100 | $100 \times K$ | 200 | $10,000$ |
| Food101 [8] | 101 | $101 \times K$ | 202 | $25,250$ |
| Flowers102 [45] | 102 | $102 \times K$ | 204 | $6,149$ |

Table 5: Dataset Details for $K$-Shot Classification problem

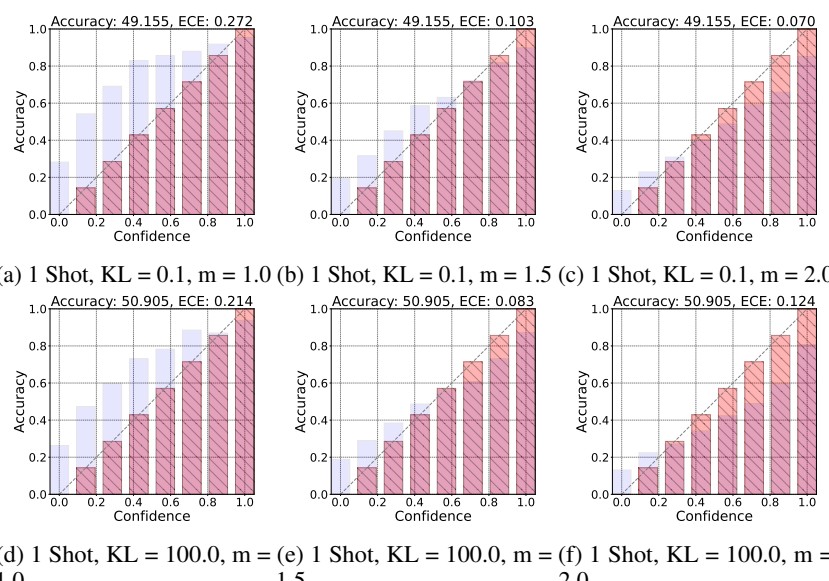

(a) 1 Shot, KL = 0.1, m = 1.0  (b) 1 Shot, KL = 0.1, m = 1.5  (c) 1 Shot, KL = 0.1, m = 2.0

(d) 1 Shot, KL = 100.0, m = 1.0  (e) 1 Shot, KL = 100.0, m = 1.5  (f) 1 Shot, KL = 100.0, m = 2.0

Figure 9: Visualization of the impact of m for 1-Shot Cifar100 dataset using reliability plots

# F  Additional Experiments

## F.1  Impact of $m$ on Expected Calibration Error

The developed B-PEFT model introduces the post-hoc calibration technique that adjusts the base rate in the evidential model with one additional hyperparameter $m$ (see Section 3.2 for details). We carry out a grid search using the few-shot validation dataset and select the optimal $m$ for the evidential models. In this section, we carry out detailed ablation to study the impact of $m$ with few-shot Cifar100 datasets. We consider $m$ values of $(0.5, 1.0, 1.5, 2.0, 2.5, 3.0)$, incorrect evidence regularization strengths of $(0.0, 0.1, 1.0, 10.0, 100.0, 1000.0)$, and few-shot values of $(1, 2, 5, 10, 20)$ (see Table 6 and Figure 9, Figure 10). Across all the experiments, we observe that as we increase $m$, the model transforms the evidence to maximize the evidence gap making the model more confident on its knowledge. With increased confidence, the model's calibration performance improves. However, a large increase in $m$ values starts to make the model overconfident leading to increased ECE and poor calibration. The strength of incorrect evidence regularization also impacts the optimal value of $m$. For large values of incorrect evidence regularization, a smaller $m$ value suffices to make the model well-calibrated. The trend is seen across all few-shot classification settings.

Table 6: Impact of m

| Reg. Strength ($\lambda$) (Eqn 2) | m = 0.5 | m = 1.0 | m = 1.5 | m = 2.0 | m = 2.5 | m = 3.0 |
|---|---|---|---|---|---|---|
| **1 Shot** | | | | | | |
| $\lambda = 0.0$ | $0.401_{\pm0.007}$ | $0.257_{\pm0.009}$ | $0.094_{\pm0.011}$ | $0.07_{\pm0.004}$ | $0.147_{\pm0.008}$ | $0.214_{\pm0.008}$ |
| $\lambda = 0.1$ | $0.423_{\pm0.008}$ | $0.261_{\pm0.007}$ | $0.092_{\pm0.008}$ | $0.079_{\pm0.003}$ | $0.156_{\pm0.007}$ | $0.22_{\pm0.008}$ |
| $\lambda = 1.0$ | $0.434_{\pm0.009}$ | $0.252_{\pm0.009}$ | $0.082_{\pm0.009}$ | $0.091_{\pm0.005}$ | $0.169_{\pm0.007}$ | $0.23_{\pm0.007}$ |
| $\lambda = 10.0$ | $0.429_{\pm0.004}$ | $0.231_{\pm0.005}$ | $0.077_{\pm0.007}$ | $0.107_{\pm0.005}$ | $0.186_{\pm0.005}$ | $0.245_{\pm0.005}$ |
| $\lambda = 100.0$ | $0.433_{\pm0.009}$ | $0.212_{\pm0.007}$ | $0.08_{\pm0.008}$ | $0.125_{\pm0.007}$ | $0.199_{\pm0.008}$ | $0.253_{\pm0.009}$ |
| $\lambda = 1000.0$ | $0.422_{\pm0.006}$ | $0.182_{\pm0.003}$ | $0.077_{\pm0.005}$ | $0.148_{\pm0.003}$ | $0.221_{\pm0.002}$ | $0.271_{\pm0.003}$ |
| **2 Shot** | | | | | | |
| $\lambda = 0.0$ | $0.533_{\pm0.005}$ | $0.334_{\pm0.009}$ | $0.120_{\pm0.011}$ | $0.041_{\pm0.002}$ | $0.100_{\pm0.009}$ | $0.154_{\pm0.008}$ |
| $\lambda = 0.1$ | $0.559_{\pm0.006}$ | $0.328_{\pm0.010}$ | $0.103_{\pm0.010}$ | $0.060_{\pm0.005}$ | $0.113_{\pm0.006}$ | $0.160_{\pm0.005}$ |
| $\lambda = 1.0$ | $0.565_{\pm0.005}$ | $0.308_{\pm0.005}$ | $0.086_{\pm0.004}$ | $0.072_{\pm0.003}$ | $0.125_{\pm0.003}$ | $0.167_{\pm0.003}$ |
| $\lambda = 10.0$ | $0.560_{\pm0.005}$ | $0.282_{\pm0.005}$ | $0.074_{\pm0.004}$ | $0.081_{\pm0.004}$ | $0.136_{\pm0.005}$ | $0.177_{\pm0.004}$ |
| $\lambda = 100.0$ | $0.550_{\pm0.002}$ | $0.264_{\pm0.003}$ | $0.066_{\pm0.002}$ | $0.087_{\pm0.003}$ | $0.146_{\pm0.002}$ | $0.186_{\pm0.002}$ |
| $\lambda = 1000.0$ | $0.547_{\pm0.002}$ | $0.257_{\pm0.001}$ | $0.064_{\pm0.001}$ | $0.090_{\pm0.002}$ | $0.147_{\pm0.002}$ | $0.186_{\pm0.002}$ |
| **5 Shot** | | | | | | |
| $\lambda = 0.0$ | $0.597_{\pm0.010}$ | $0.354_{\pm0.017}$ | $0.125_{\pm0.017}$ | $0.031_{\pm0.004}$ | $0.073_{\pm0.011}$ | $0.117_{\pm0.010}$ |
| $\lambda = 0.1$ | $0.639_{\pm0.007}$ | $0.349_{\pm0.007}$ | $0.109_{\pm0.005}$ | $0.042_{\pm0.004}$ | $0.084_{\pm0.002}$ | $0.119_{\pm0.003}$ |
| $\lambda = 1.0$ | $0.634_{\pm0.002}$ | $0.340_{\pm0.003}$ | $0.111_{\pm0.002}$ | $0.047_{\pm0.006}$ | $0.085_{\pm0.003}$ | $0.122_{\pm0.002}$ |
| $\lambda = 10.0$ | $0.645_{\pm0.004}$ | $0.362_{\pm0.005}$ | $0.138_{\pm0.003}$ | $0.049_{\pm0.004}$ | $0.071_{\pm0.001}$ | $0.107_{\pm0.002}$ |
| $\lambda = 100.0$ | $0.652_{\pm0.006}$ | $0.357_{\pm0.001}$ | $0.119_{\pm0.001}$ | $0.042_{\pm0.002}$ | $0.075_{\pm0.003}$ | $0.109_{\pm0.003}$ |
| $\lambda = 1000.0$ | $0.642_{\pm0.006}$ | $0.314_{\pm0.003}$ | $0.081_{\pm0.002}$ | $0.051_{\pm0.002}$ | $0.091_{\pm0.004}$ | $0.122_{\pm0.005}$ |
| **10 Shot** | | | | | | |
| $\lambda = 0.0$ | $0.604_{\pm0.017}$ | $0.325_{\pm0.007}$ | $0.090_{\pm0.004}$ | $0.041_{\pm0.003}$ | $0.095_{\pm0.005}$ | $0.134_{\pm0.007}$ |
| $\lambda = 0.1$ | $0.629_{\pm0.009}$ | $0.314_{\pm0.005}$ | $0.077_{\pm0.005}$ | $0.054_{\pm0.004}$ | $0.100_{\pm0.004}$ | $0.134_{\pm0.004}$ |
| $\lambda = 1.0$ | $0.657_{\pm0.006}$ | $0.367_{\pm0.003}$ | $0.136_{\pm0.002}$ | $0.053_{\pm0.003}$ | $0.068_{\pm0.003}$ | $0.102_{\pm0.004}$ |
| $\lambda = 10.0$ | $0.685_{\pm0.004}$ | $0.404_{\pm0.002}$ | $0.151_{\pm0.001}$ | $0.034_{\pm0.002}$ | $0.057_{\pm0.001}$ | $0.090_{\pm0.002}$ |
| $\lambda = 100.0$ | $0.673_{\pm0.007}$ | $0.363_{\pm0.004}$ | $0.109_{\pm0.004}$ | $0.041_{\pm0.001}$ | $0.075_{\pm0.001}$ | $0.104_{\pm0.003}$ |
| $\lambda = 1000.0$ | $0.651_{\pm0.004}$ | $0.335_{\pm0.005}$ | $0.099_{\pm0.006}$ | $0.049_{\pm0.001}$ | $0.082_{\pm0.001}$ | $0.113_{\pm0.001}$ |
| **20 Shot** | | | | | | |
| $\lambda = 0.0$ | $0.413_{\pm0.003}$ | $0.141_{\pm0.008}$ | $0.060_{\pm0.010}$ | $0.162_{\pm0.011}$ | $0.226_{\pm0.011}$ | $0.270_{\pm0.010}$ |
| $\lambda = 0.1$ | $0.409_{\pm0.002}$ | $0.099_{\pm0.005}$ | $0.090_{\pm0.006}$ | $0.188_{\pm0.007}$ | $0.248_{\pm0.006}$ | $0.288_{\pm0.006}$ |
| $\lambda = 1.0$ | $0.399_{\pm0.002}$ | $0.091_{\pm0.002}$ | $0.086_{\pm0.002}$ | $0.183_{\pm0.002}$ | $0.244_{\pm0.002}$ | $0.284_{\pm0.002}$ |
| $\lambda = 10.0$ | $0.387_{\pm0.002}$ | $0.075_{\pm0.004}$ | $0.106_{\pm0.003}$ | $0.202_{\pm0.003}$ | $0.259_{\pm0.002}$ | $0.297_{\pm0.002}$ |
| $\lambda = 100.0$ | $0.401_{\pm0.012}$ | $0.097_{\pm0.024}$ | $0.093_{\pm0.021}$ | $0.196_{\pm0.016}$ | $0.259_{\pm0.012}$ | $0.299_{\pm0.009}$ |

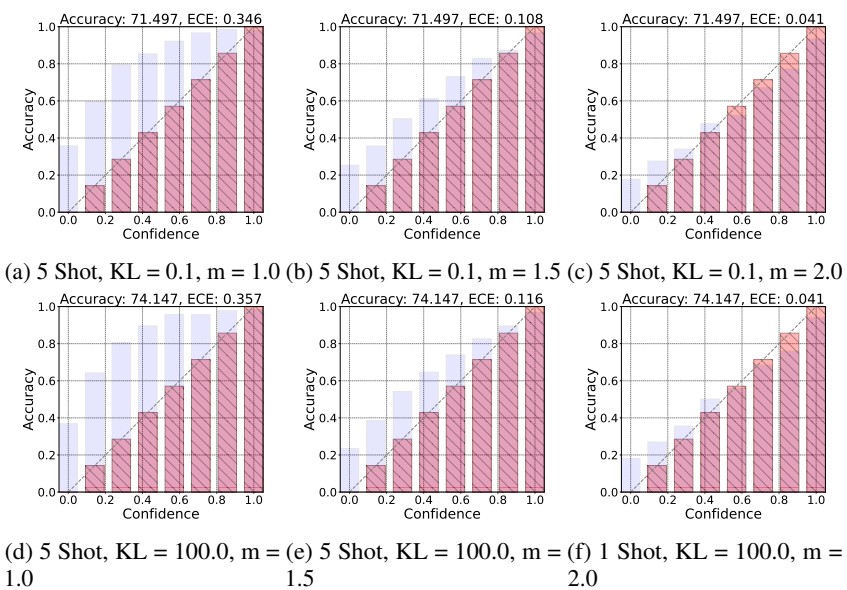

(a) 5 Shot, KL = 0.1, m = 1.0 (b) 5 Shot, KL = 0.1, m = 1.5 (c) 5 Shot, KL = 0.1, m = 2.0

(d) 5 Shot, KL = 100.0, m = (e) 5 Shot, KL = 100.0, m = (f) 1 Shot, KL = 100.0, m =
1.0                              1.5                              2.0

Figure 10: Visualization of the impact of m for 5-Shot Cifar100 dataset using reliability plots

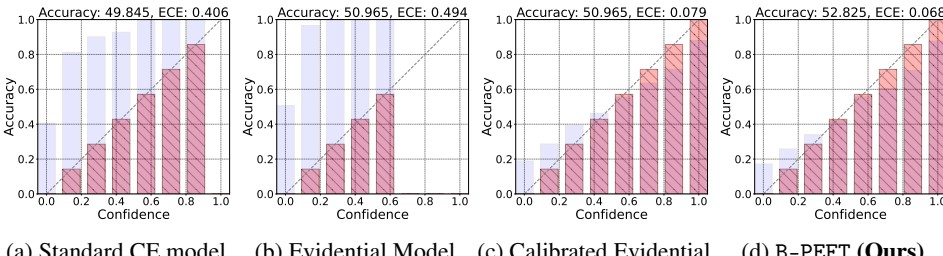

(a) Standard CE model     (b) Evidential Model     (c) Calibrated Evidential     (d) B-PEFT **(Ours)**

Figure 11: Accuracy-Confidence trends in 1-shot Cifar100 Results

## F.2 Impact of Incorrect Evidence Regularization Strength ($\lambda$)

Evidential deep learning models introduce incorrect evidence regularization to minimize the evidence of classes other than the ground truth class. In this work, we use KL divergence-based incorrect evidence regularization (see Section C.2), and introduce a hyperparameter $\lambda$ that controls the priority the model places on minimizing the incorrect class evidence over maximizing the correct class evidence. In this section, we study the impact of the hyperparameter $\lambda$ on the model performance with $K$-shot Cifar100 and Flowers102 experiments (We experiment with $K$ values of $(1, 2, 5, 10, 20)$). We observe that the model's calibration performance (the ECE) is optimal when no incorrect evidence regularization is used *i.e.,* $\lambda = 0$ (see Table 7 and Table 8). However, no incorrect evidence regularization hurts the model's generalization performance. With the increase in incorrect evidence regularization, the model's generalization performance (indicated by accuracy) improves albeit with ECE tradeoff. However, very large incorrect evidence regularization misguides the model to only focus on minimizing incorrect class evidence hurting generalization. The optimal $\lambda$ value leads to the best generalization performance while hurting the ECE performance. Moreover, with a larger number of shots in training, the optimal $\lambda$ value is generally smaller (For instance, in 1-shot Cifar100, optimal $\lambda = 1000.0$, in 2-shot Cifar100, optimal $\lambda = 10.0$, and in 5-shot Cifar100, optimal $\lambda = 0.1$). Once the optimal $\lambda$ value is determined, we can resort to our evidential base-rate adjustment that leads to a calibrated evidential model with good generalization performance.

Table 7: Different Shot Classification – Accuracy and ECE in Cifar100

| Shots | KL 0.0 | KL 0.1 | KL 1.0 | KL 10.0 | KL 100.0 | KL 1000.0 |
|---|---|---|---|---|---|---|
| **(a) Accuracy ↑** | | | | | | |
| 1 | $45.502_{\pm 0.492}$ | $47.707_{\pm 0.727}$ | $48.754_{\pm 0.672}$ | $49.967_{\pm 0.569}$ | $50.127_{\pm 0.962}$ | $51.127_{\pm 0.435}$ |
| 2 | $60.494_{\pm 1.567}$ | $64.006_{\pm 0.834}$ | $64.820_{\pm 0.586}$ | $65.545_{\pm 0.339}$ | $65.439_{\pm 0.295}$ | $65.531_{\pm 0.301}$ |
| 5 | $74.230_{\pm 1.099}$ | $77.391_{\pm 1.053}$ | $77.238_{\pm 0.694}$ | $76.760_{\pm 0.652}$ | $77.561_{\pm 0.716}$ | $77.525_{\pm 0.685}$ |
| 10 | $80.566_{\pm 0.368}$ | $81.512_{\pm 0.163}$ | $81.055_{\pm 0.185}$ | $81.561_{\pm 0.291}$ | $81.559_{\pm 0.253}$ | $81.344_{\pm 0.2}$ |
| 20 | $82.111_{\pm 0.684}$ | $82.967_{\pm 0.2}$ | $83.012_{\pm 0.123}$ | $83.100_{\pm 0.184}$ | $83.014_{\pm 0.103}$ | $81.966_{\pm 0.7}$ |
| **(b) ECE ↓** | | | | | | |
| 1 | $0.404_{\pm 0.005}$ | $0.440_{\pm 0.006}$ | $0.460_{\pm 0.007}$ | $0.479_{\pm 0.005}$ | $0.487_{\pm 0.009}$ | $0.499_{\pm 0.004}$ |
| 2 | $0.480_{\pm 0.010}$ | $0.562_{\pm 0.007}$ | $0.596_{\pm 0.005}$ | $0.620_{\pm 0.004}$ | $0.631_{\pm 0.003}$ | $0.637_{\pm 0.003}$ |
| 5 | $0.513_{\pm 0.006}$ | $0.632_{\pm 0.007}$ | $0.686_{\pm 0.003}$ | $0.715_{\pm 0.005}$ | $0.744_{\pm 0.006}$ | $0.756_{\pm 0.006}$ |
| 10 | $0.499_{\pm 0.004}$ | $0.644_{\pm 0.004}$ | $0.712_{\pm 0.001}$ | $0.765_{\pm 0.002}$ | $0.787_{\pm 0.002}$ | $0.740_{\pm 0.007}$ |
| 20 | $0.483_{\pm 0.003}$ | $0.647_{\pm 0.001}$ | $0.733_{\pm 0.001}$ | $0.782_{\pm 0.001}$ | $0.804_{\pm 0.001}$ | $0.490_{\pm 0.007}$ |

Table 8: Different Shot Classification – Accuracy and ECE in Flowers102

| Shots | KL 0.0 | KL 0.1 | KL 1.0 | KL 10.0 | KL 100.0 | KL 1000.0 |
|---|---|---|---|---|---|---|
| **(a) Accuracy ↑** | | | | | | |
| 1 Shot | $84.314_{\pm 1.571}$ | $86.434_{\pm 0.437}$ | $88.481_{\pm 0.92}$ | $89.225_{\pm 1.03}$ | $89.475_{\pm 1.045}$ | $89.882_{\pm 0.592}$ |
| 2 Shot | $91.416_{\pm 1.296}$ | $93.795_{\pm 0.557}$ | $94.575_{\pm 0.504}$ | $94.899_{\pm 0.209}$ | $94.8_{\pm 0.409}$ | $95.071_{\pm 0.413}$ |
| 5 Shot | $97.471_{\pm 0.135}$ | $97.51_{\pm 0.279}$ | $97.602_{\pm 0.199}$ | $97.139_{\pm 0.185}$ | $96.953_{\pm 0.19}$ | $97.235_{\pm 0.248}$ |
| 10 Shot | $98.326_{\pm 0.233}$ | $97.964_{\pm 0.198}$ | $97.804_{\pm 0.093}$ | $97.847_{\pm 0.064}$ | $98.093_{\pm 0.14}$ | $98.034_{\pm 0.111}$ |
| 20 Shot | $98.708_{\pm 0.014}$ | $98.37_{\pm 0.072}$ | $97.953_{\pm 0.115}$ | $98.086_{\pm 0.176}$ | $98.406_{\pm 0.167}$ | $97.75_{\pm 0.804}$ |
| **(b) ECE ↓** | | | | | | |
| 1 Shot | $0.662_{\pm 0.015}$ | $0.722_{\pm 0.008}$ | $0.766_{\pm 0.008}$ | $0.801_{\pm 0.008}$ | $0.826_{\pm 0.009}$ | $0.846_{\pm 0.004}$ |
| 2 Shot | $0.608_{\pm 0.005}$ | $0.696_{\pm 0.003}$ | $0.747_{\pm 0.008}$ | $0.794_{\pm 0.008}$ | $0.835_{\pm 0.009}$ | $0.874_{\pm 0.006}$ |
| 5 Shot | $0.487_{\pm 0.014}$ | $0.598_{\pm 0.013}$ | $0.686_{\pm 0.02}$ | $0.731_{\pm 0.005}$ | $0.83_{\pm 0.019}$ | $0.896_{\pm 0.004}$ |
| 10 Shot | $0.444_{\pm 0.008}$ | $0.57_{\pm 0.026}$ | $0.641_{\pm 0.008}$ | $0.733_{\pm 0.013}$ | $0.827_{\pm 0.018}$ | $0.908_{\pm 0.01}$ |
| 20 Shot | $0.411_{\pm 0.013}$ | $0.544_{\pm 0.009}$ | $0.634_{\pm 0.012}$ | $0.75_{\pm 0.046}$ | $0.831_{\pm 0.023}$ | $0.898_{\pm 0.007}$ |

### F.3 Impact of Ensemble Components

We present the experiment to study the impact of ensemble components in Figure 12. The experiment is performed on 1 one-shot Cifar100 dataset using prompt-based adaption for vision transformer. There is a performance gain of both accuracy and ECE with the use of all ensemble components. We also note that as we increase the number of components from 1 to 3, both the generalization and calibration performance increase significantly. For instance, the ECE with a single ensemble component is 0.088 which improves to 0.077 with 3 ensemble components while the accuracy improves by almost 3%.. However, with a further increase in the number of ensemble components, the generalization/calibration performance

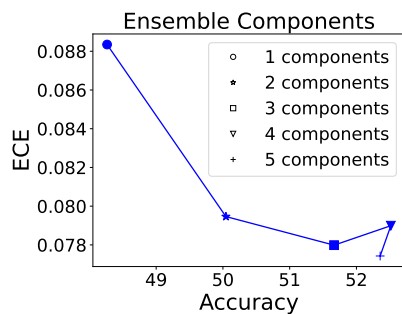

Figure 12: Impact of ensemble components as number of component increases

improvement is not significant. Thus, for our B-PEFT model, we carry out an ensemble of 3 evidential models that is a good balance between the number of ensemble components and the gain in generalization/calibration performance.

### F.4 Few Shot Learning Results

In this set of experiments, we apply our model to the 5-way 1-shot mini-ImageNet testing tasks and compare it with representative meta-learning models. The results are summarized in Table 9. Benefiting from the knowledge acquired during pre-training, the proposed PEFT-based model outperforms the episodic meta-learning models by a large margin, demonstrating the potential of large vision foundation models for effective few-shot learning. However, the VPT model is under-confident as shown in Table 10 and Figure 13. Our B-PEFT model improves on the VPT model's generalization and calibration leading to promising few-shot adaptation results.

Table 9: 5-Way 1-Shot Mini-ImageNet

| Model | Accuracy |
|---|---|
| MAML [16] | 48.70 |
| Matching Networks [64] | 43.56 |
| LLAMA [24] | 49.40 |
| VERSA [23] | 53.40 |
| PLATIPUS [17] | 50.13 |
| Bayesian-MAML [69] | 53.30 |
| Bayesian-TAML [37] | 71.46 |
| VPT | 89.50 |
| **B-PEFT (Ours)** | **90.09** |

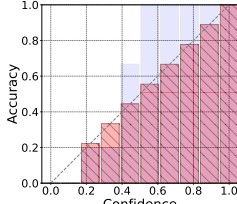

Figure 13: VPT reliability plot for 5-Way 1-Shot Mini-ImageNet

Table 10: Calibration Results

| Model | Accuracy | ECE |
|---|---|---|
| VPT | 89.50 | 0.418 |
| **B-PEFT (Ours)** | **90.09** | **0.065** |

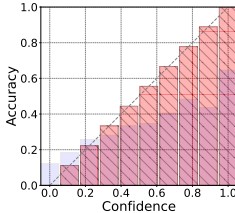

Figure 14: Full fine tuning reliability plot for 100-way 1-Shot Cifar100

### F.5 Impact of Different Components

In this section, we study the under-confidence behavior of PEFT methods w.r.t. data size, number of classes, and number of unfrozen parameters.

**Data size.** We observe that the model's accuracy increases with more training samples (see Table 1 where we vary shots from 1 to 20). The under-confidence remains, even with further increase in training samples. To this end, we conduct additional experiments on Cifar100 by increasing the training samples per class to 500 and observe the under-confidence issue despite the increase in the accuracy. The trend is summarized in Figure 15 (a-b). We see an increase in accuracy and a decrease in ECE. However, even with 500 samples per class, the under-confidence issue remains. Further, we report the accuracy and ECE of the fully fine-tuned model (fine-tuning of all the parameters) for 1 shot cifar100 in Table 11 where we observe a decrease in accuracy while the calibration issue remains. We observe that full fine-tuning leads to overconfidence behavior, hurting the generalization performance, as seen in Table 11 and reliability plot as presented in Figure 14.

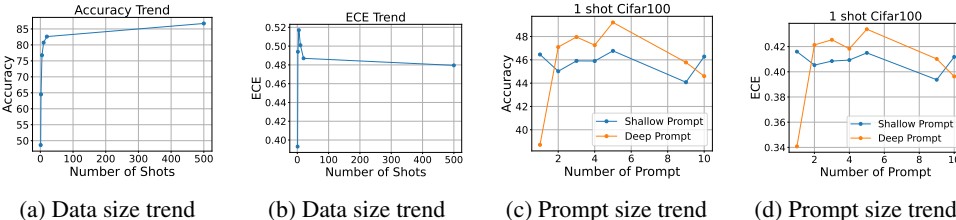

| (a) Data size trend | (b) Data size trend | (c) Prompt size trend | (d) Prompt size trend |

Figure 15: (a-b) Accuracy-EEC trend with the number of parameters, (c-d): Accuracy-ECE Trend with data size

**Number of classes:** To study the impact of number of classes, we formulate 5-way 1 shot, 10-way 1 shot, and 100-way 1 shot tasks using Cifar100. The results are presented in Table 12. As we decrease the number of shots from 100 to 5, we see an increase in accuracy and a decrease in ECE. We observe that the model is more accurate as tasks become easier (indicated by fewer classes *i.e.,* lower $N$ value in Table 12). However, the under-confidence issue remains.

**Number of unfrozen parameters:** We conduct additional experiments on Cifar100 100-way 1-shot tasks by varying the number of prompts for 1) shallow prompt: prompt added to the input only and 2) deep prompt: prompt added to all Transformer encoder layers' input as well. The accuracy and ECE trends are presented in Figure 15 (c-d). As can be seen, with the increase in the number of prompts for both shallow and deep prompts, there are fluctuations in accuracy and ECE performance. However, the under-confidence issue persists for all the cases.

## F.6  Additional Experiments and Comparison

In this section, we carry out additional experiments to study the OOD performance of our model, and compare our model with additional methods present in literature.

**OOD Performance:**  The current work, being an instance of fine-grained uncertainty quantification works, could potentially help in OOD detection. To this end, we present the OOD results of Cifar10 as in-distribution dataset and Cifar100 as out-of-distribution dataset with AUROC, FPR95, AUPR metrics for our model on 100-way 1-shot and 100-way 5-shot Cifar100 tasks in Table 13. As seen, B-PEFT performs better than PEFT, and with more training data, the model's OOD detection capabilities improve. Even with only 5 samples/class (i.e. 100-way 5-shot Cifar100 task), the model can achieve an AUROC of 92.58.

**Model Comparison:**  We first carry out experiments with cosine classifier [22] without training for the 100-way 1-shot task on Cifar100. The cosine classifier has comparable generalization performance (accuracy) in comparison to VPT based model (see Table 14). However, looking at the ECE, the miscalibration issue is even higher than VPT based model. Hence, the simple solution (cosine classifier) does not ensure calibrated predictions. We also carry out experiments with test time augmentation [4] and VPT fine tuning with LoRA [70], on 100-way 1-shot Cifar100 dataset. Towards comparison with Bayesian inspired methods [14], we use Laplace approximation on last layer of the model using Kronecker Product and Diagonalization represented by KronLaplace and DiagLaplace in Table 14. As can be seen, these methods also suffer from the under-confidence issue when straightforwardly extended to the VPT. B-PEFT achieves much better generalization and calibration performance than these baselines.

Table 11: Full fine-tuning results

| Model | Accuracy | ECE |
|-------|----------|-----|
| Full Fine Tuning | 25.75 | 0.118 |
| FT + Base rate adj. | 25.75 | 0.037 |
| VPT | 48.63 | 0.393 |
| **B-PEFT** | **52.34** | **0.067** |

Table 12: N-Way 1-Shot Cifar100 calibration

| Task | Accuracy | ECE |
|------|----------|-----|
| 5 way 1 shot | 63.60 | 0.324 |
| 10 way 1 shot | 56.35 | 0.370 |
| 100 way 1 shot | 48.63 | 0.393 |

Table 13: 100-way Cifar100 OOD experiments

| PEFT | AUROC | AUPR | FPR95 |
|------|-------|------|-------|
| 1 shot | 79.53 | 80.09 | 70.58 |
| 5 shot | 90.93 | 90.75 | 39.82 |
| **(B-PEFT)** | **AUROC** | **AUPR** | **FPR95** |
| 1 shot | 81.24 | 81.98 | 68.15 |
| 5 shot | 92.58 | 92.85 | 35.24 |

Table 14: Comparison with baselines

| Model | Accuracy | ECE |
|-------|----------|-----|
| Cosine Classifier | 47.99 | 0.493 |
| ViT + LoRA | 48.19 | 0.243 |
| Test Time Aug. | 50.10 | 0.271 |
| VPT + KronLaplace | 50.26 | 0.475 |
| VPT + DiagLaplace | 50.20 | 0.474 |
| VPT | 48.63 | 0.393 |
| **B-PEFT** | **52.34** | **0.067** |

# G   Calibration Behavior of Self-Supervised Model

In this work, we focus on vision foundation models that are pre-trained in a supervised learning paradigm. These models have shown remarkable effectiveness in a wide range of areas including image classification, video understanding, and visual recognition among others. Alternatively, foundation models have been developed that pre-train in a self-supervised fashion (*e.g.,* CLIP [52]). These models demonstrate good zero-shot performance on various datasets. As a representative of the self-supervised models, we use CLIP in our experiments. Parameter-efficient methods [77, 76, 21] have been proposed to adapt the CLIP model for downstream tasks. We use the popular methods: adapter and prompt for few-shot adaptation to study the calibration behavior. The reliability plot of different shot adaptations (1,2 and 5

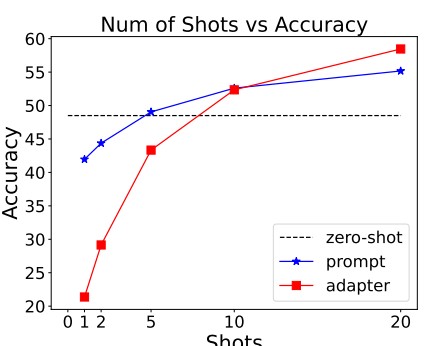

Figure 16: Accuracy trends of CLIP on few-shot adaptation

shots per class) for cifar100 along with accuracy and ECE is presented in Figure 17. The accuracy behavior as we increase the number of shots from 1 to 20 is shown in Figure 16. In both methods, we observe for few-shot adaptations, the accuracy is either lower or comparable to zero-shot performance. More specifically, for adapter-based adaptation, even with 5 shots per class, the accuracy does not reach zero-shot accuracy. On the contrary, the parameter-efficient fine-tuning of supervised foundation models shows consistent improvement in accuracy as we increase the number of shots.

Towards calibration performance, we observe that prompt-based adaptation, along with zero-shot generally show good calibration performance with some degree of overfitting (see Figure 17). In contrast, prompt adaptation for supervised models that are severe, and prompt adaptation for CLIP models have no such issue. However, the adapter-based adaptation is mostly over-confident and hence has a higher ECE value than our proposed model. Considering these results for parameter-efficient few-shot adaption of pre-trained self-supervised models, the calibration and uncertainty behavior of such pre-trained models pose an interesting direction for further investigation.

# H   Societal Impact

We study the parameter-efficient fine-tuning techniques for large pre-trained vision foundation models, where we identify two key issues: the under-confidence of the fine-tuned models in their

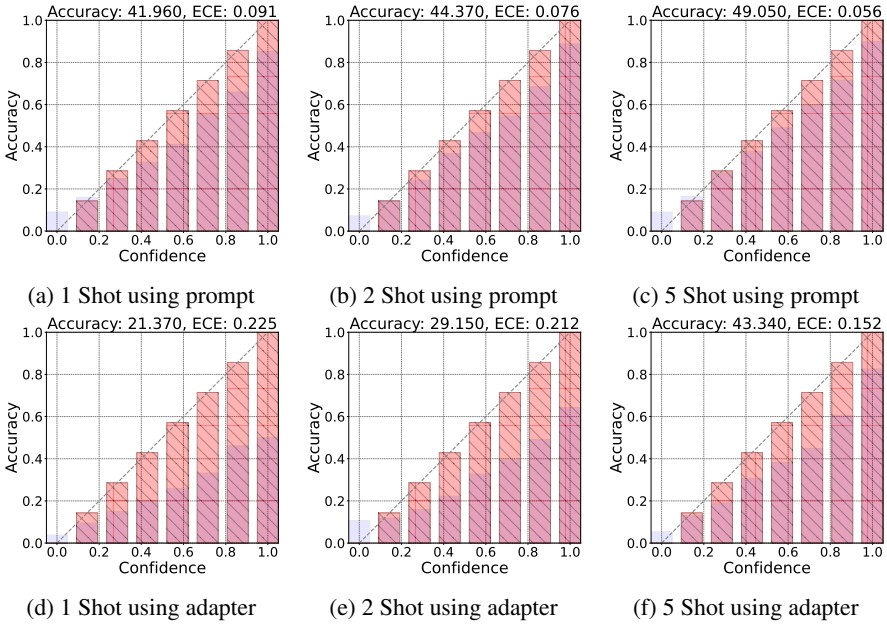

Figure 17: Different Shot adaptation: Calibration performance of CLIP based model on cifar100 dataset using prompt and adapter-based fine-tuning

predictions, and lack of fine-grained uncertainty quantification capabilities. We develop a novel Bayesian Evidential model: B-PEFT that addresses the weaknesses of existing PEFT for pre-trained foundation models. Being an instance of the PEFT, our developed model enables the large foundation models to be adapted to challenging few-shot problems in a parameter-efficient and computationally efficient manner with limited memory requirements and energy footprint. Moreover, the developed model improves the generalization performance and the model's predictions are calibrated ensuring trustworthiness. Finally, the model has fine-grained uncertainty quantification capabilities which are highly desirable when applying these models in real-world safety-critical scenarios. Overall, our developed B-PEFT is expected to have a strong positive societal impact.

# I   Limitations and Future Work

In this work, we investigate the calibration of transformer-based foundation models under few-shot adaptation using various parameter-efficient fine-tuning methods. We focus on fine-tuning supervised pre-trained models for few-shot learning. We note that there are other self-supervised pre-trained models that show promising results for various benchmark datasets. We investigate the calibration of CLIP, a representative method, under few-shot adaptation using the two most popular parameter-efficient fine-tuning methods: prompt and adapter. Our preliminary results demonstrate that the few-shot performance does not consistently increase in comparison to zero-shot. Similarly, prompt-based fine-tuning has relatively better calibration than adapter-based fine-tuning. As an extension of this work, we will investigate the calibration performance of self-supervised foundation models. Additionally, it could be interesting to study the calibration performance of PEFT for tasks beyond image classification, *e.g.,* to other modalities such as audio and language foundation models. For instance, the ideas developed in this work could potentially be used in situations where the PEFT leads to mis-calibrated models and the developed model requires trustworthy fine-grained uncertainty quantification capabilities. If these data modalities are also modeled using transformers and follow the parameter-efficient fine-tuning paradigm in performing downstream tasks, we expect the proposed approach can benefit them in a similar way.

