# OpenReview forum: "Be Confident in What You Know: Bayesian Parameter Efficient Fine-Tuning of Vision Foundation Models"
_NeurIPS.cc/2024/Conference — NeurIPS 2024 poster_

### Official Review · Reviewer_CbTK · 2024-07-10

**Soundness:** 3
**Presentation:** 3
**Contribution:** 2
**Rating:** 6
**Confidence:** 4

**Summary:**

This paper focuses on the parameter efficient fine-tuning (PEFT) for foundation models in the few-shot learning settings. The authors reveal that the adapted models lack accurate fine-grained uncertainty quantification capabilities. Specifically, the few-shot tuned models perform remarkable accuracy but with large expected calibration errors. To address this problem, they propose a lightweight Bayesian PEFT framework by designing two Bayesian components to improve models' confidence. Theoretical analysis and experiments on 4 visual benchmarks prove the effectiveness of the proposed method.

**Strengths:**

1.The paper is well-written and the storyline is clear.
2.The proposed Bayesian-PEFT is simple-and-effective, and theoretical analysis in the paper can reveal the underlying mechanism of the method.
3.Extensive experiments on ViT in several few-shot settings show the effectiveness of Bayesian-PEFT.

**Weaknesses:**

1.The title may be a bit inappropriate. If the Bayesian-PEFT is claimed to be used for foundation models, experiments on more types of foundation models are required. For example, I am curious whether it still works well on large language models (LLMs), e.g., LLaMA3. Also, it is unclear whether the under-confidence issue observed in the ViT also exists in LLMs.
2.Although it may be beyond the scope of the paper's research, I am still curious whether the under-confidence issue studied in the paper disappears when there is sufficient training data.
3.It would be better if a schematic diagram of the proposed Bayesian-PEFT could be provided.

**Questions:**

Does the proposed Bayesian-PEFT work well in LLMs, e.g., LLaMA3?

**Limitations:**

Yes

---

> ### Author Rebuttal · Authors · 2024-08-07
>
> Thank you for taking the time to review our paper. We appreciate your constructive comments and suggestions.  Here are the responses to the clarifying questions:
>
> **Q1: LLM Clarification**
>
>
> The proposed ideas in this work could potentially be extended to PEFT of LLM models for some downstream tasks. However, it is worth noting that the uncertainty quantification for LLM is challenging due to their generative nature, and is an open question. For instance, LLAMA3 is a generative model, and uncertainty quantification of the overall generated text response for an input query is difficult to interpret. In this work, we focus on PEFT of vision foundation models, where we identify the under-confidence issue for all 4 PEFT techniques considered. We introduce a novel evidence-guided method to address this challenge. We will update the title from foundation models to vision foundation models to make it more accurate.
>
> **Q2: Under-confidence with sufficient data**
>
> Please refer to the answer to **Q1** in the general response for the results related to PEFT with sufficient data.
>
>
> **Q3: Schematic Diagram of B-PEFT**
>
>  We present the schematic diagram of B-PEFT in Figure 3 of the attached pdf of the general response. We will include the schematic diagram in the revised draft.

---

> > ### Author Response · Authors · 2024-08-11
> > **Official Comment by Authors**
> >
> > Dear Reviewer CbTK,
> >
> > We’d like to express our gratitude again towards the reviewer for evaluating the proposed work and providing constructive comments. In our rebuttal,
> >
> > - We analyze under-confidence behavior with varying sizes of data.
> >
> > - We provide the schematic diagram of B-PEFT method.
> >
> > - We discuss the potential extension to LLM models.
> >
> > We are happy to answer any further questions that you may have regarding our work.

---

### Official Review · Reviewer_YJfL · 2024-07-14

**Soundness:** 3
**Presentation:** 3
**Contribution:** 3
**Rating:** 6
**Confidence:** 2

**Summary:**

This submission presented a lightweight Bayesian Parameter Efficient Fine-Tuning (Bayesian-PEFT) framework for large transformer-based foundation models. Experiments across diverse datasets demonstrated the improved calibration performance by Bayesian-PEFT on multiple other PEFT techniques.

**Strengths:**

* The paper is well written and the experiments are extensive.

* The proposed transformation function $A_m$ is well motivated and the experiments supports authors' claim.

**Weaknesses:**

* The dependence on Evidential Deep Learning is quite significant. To some extent, the proposed method extends EDL, rather than being designed specifically for PEFT of foundation models.

* Some editorial errors (e.g., "we we") exists, the submission may need further proofreading.

**Questions:**

* In Eq. (6), would there be a numerical stability problem when dividing e_{min}, as this value can be very small?

**Limitations:**

Yes.

---

> ### Author Rebuttal · Authors · 2024-08-07
>
> Thank you for taking the time to review our paper. We appreciate your constructive comments and suggestions.  Here are the responses to the clarifying questions:
>
> **Q1: The dependence on Evidential Deep Learning is quite significant. To some extent, the proposed method extends EDL, rather than being designed specifically for PEFT of foundation models.**
>
> Please refer to the answer to **Q2** in the general response, which makes it clear that the proposed approach is specifically designed to address the unique under-confidence behavior of PEFT methods. We extend evidential  learning in novel ways by performing fine-grained uncertainty decomposition and base rate adjustment, which allows us to seamlessly integrate it into parameter efficient fine-tuning paradigm. Such an integration has the potential to significantly broaden the usage of  foundation models in many critical domains by fixing their calibration issues.
>
> **Q2: Typos and minor mistakes**
>
> We will proofread the draft, and correct typos/minor mistakes in the updated draft.
>
> **Q3: Numerical Stability Issue in Eqn. 6**
>
> Yes, since the evidence is in the range [0, infinity] (the evidence is obtained by the exponential of the logit outputs), the stability issue can arise in extreme cases when the logit output is extremely low (i.e. close to negative infinity). In our experiments, we did not observe the stability issue. Still, the issue can arise in some extreme cases, and to address this challenge, we could introduce a small delta in the denominator or bound the network’s logits to be greater than a small negative value.

---

> > ### Author Response · Authors · 2024-08-11
> > **Official Comment by Authors**
> >
> > Dear Reviewer YJfL,
> >
> > We’d like to express our gratitude again towards the reviewer for evaluating the proposed work and providing constructive comments. In our rebuttal,
> >
> > - We emphasize the novelty of the proposed method to address the unique under-confidence behavior of the PEFT methods.
> >
> > - We clarify the numerical instability issue that can arise during experiments and how to tackle the issue.
> >
> > We are happy to answer any further questions that you may have regarding our work.

---

### Official Review · Reviewer_DMeg · 2024-07-16

**Soundness:** 3
**Presentation:** 3
**Contribution:** 3
**Rating:** 6
**Confidence:** 3

**Summary:**

This paper studies the problem of parameter efficient fine-tuning. The authors pointed out mis-calibration issues caused by parameter efficient fine-tuning, which is, more specifically, under-confident estimation when fine-tuning data is limited. To solve this issue, the author proposed Bayesian PEFT, a Bayesian parameter efficient fine-tuning method that introduces two components, i.e., a base rate adjustment to strengthen the prior belief of pre-trained knowledge, and a evidential ensemble with diverse ensemble components. The authors provided both theoretical justification as well as extensive experiments across diverse datasets.

**Strengths:**

1. This paper is well written and easy to follow.

2. The paper discusses an interesting observation on low confidence in few-shot setup for PEFT.

3. The proposed method seems to be reasonable and solve the identified problem.

**Weaknesses:**

1. It would be interesting to see more in depth discussion and empirical analysis on the observation of uncertainty of PEFT, particularly, related to data size and number of unfrozen parameters.

2. More discussion related to OOD generalization might be needed. Particularly, how does the proposed method help OOD generalization in terms of prediction accuracy.

**Questions:**

1. Is fully fine-tuned model also have low uncertainty compared to PEFT model?

2. Will the proposed method improve robustness such as OOD generalization in terms of model accuracy?

3. How does the proposed approach work on other modalities, such as audio and language?

**Limitations:**

Yes.

---

> ### Author Rebuttal · Authors · 2024-08-07
>
> Thank you for taking the time to review our paper. We appreciate your constructive comments and suggestions.  Here are the responses to the clarifying questions:
>
> **Q1: It would be interesting to see more in depth discussion and empirical analysis on the observation of uncertainty of PEFT, particularly, related to data size and number of unfrozen parameters.**
>
> Please see the answer to **Q1** in the overall response.
>
> **Q2: Clarification on OOD performance**
>
> The current work, being an instance of fine-grained uncertainty quantification works, can help in OOD detection. We further present the OOD performance of our model in the answer to **Q3** of the overall response, that shows the potential of our model in handling OOD.
>
> **Q3: Is fully fine-tuned model also have low uncertainty compared to PEFT model?**
>
> Full fine-tuning in few-shot learning tasks leads to overconfident models (see Table 5/Figure 1(a) in the attached pdf) and, due to overfitting, also hurts the model's generalization performance.
>
> **Q4: How does the proposed approach work on other modalities, such as audio and language?**
>
> In this work, we focus on few-shot image classification task and extension to other modalities is beyond the scope of this work.
> It could be interesting follow-up to study the calibration performance of PEFT for audio/language foundation models. For instance, the ideas developed in this work could potentially be used in situations where the PEFT leads to miscalibrated models and the developed model requires trustworthy fine-grained uncertainty quantification capabilities. If these data modalities are also modeled using transformers and follow the parameter efficient fine-tuning paradigm in performing downstream tasks, we expect the proposed approach can benefit them in a similar way.

---

> > ### Author Response · Authors · 2024-08-11
> > **Official Comment by Authors**
> >
> > Dear Reviewer DMeg,
> >
> > We’d like to express our gratitude again towards the reviewer for evaluating the proposed work and providing constructive comments. In our rebuttal,
> >
> > - We conduct an in-depth analysis on the under-confidence behavior of PEFT methods w.r.t number of classes, data size, and the number of unfrozen parameters.
> >
> > - We fully fine-tune the model to analyse the calibration behavior.
> >
> > - We discuss the possible extension of the proposed method to other modalities, such as audio and language.
> >
> > We are happy to answer any further questions that you may have regarding our work.

---

### Official Review · Reviewer_7WZR · 2024-07-16

**Soundness:** 2
**Presentation:** 3
**Contribution:** 2
**Rating:** 6
**Confidence:** 3

**Summary:**

This paper advances the Bayesian Parameter Efficient Fine-Tuning (Bayesian-PEFT) approach: a strategy to fine-tune large pre-trained foundation models with well calibrated classifier and ability to gracefully deal with out-of-distribution (OOD) data thanks to uncertainty quantification from ensembling.
First the authors study the behavior of a fine-tuned foundation model with several PEFT methods (bias fine-tuning, adapters, side-tuning, visual prompt-tuning) in the few-shot learning regime (defined here as reducing the number of training samples per class, but keeping the original number of classes) and show that even though they achieve good accuracy, they all display under-confident predictions. The authors attempt to trace back the source of this behavior with evidential learning tools and conclude that the models are accurate thanks to relatively greater evidence of the correct class compared to other classes, and that models are under-confident because all classes are being assigned very low evidence, including the correct ones.
Inspired by this finding, the authors devise an evidential learning based PEFT strategy that adjusts the base rates of the evidential head such that prior belief from pre-training knowledge is strengthened while maintaining accuracy. Secondly the authors propose an evidential ensemble with a technique to induce diversity with different incorrect belief regularization strengths that penalize the model differently when assigning high beliefs to non-correct classes.
The proposed Bayesian-PEFT is evaluated on 4 datasets  (CIFAR10, CIFAR100, Food101, Flowers102) under different few-shot regimes, showing boosted accuracy and calibration.

**Strengths:**

### Significance
- this work deals with an important problem in the current landscape of machine learning: how to leverage existing foundation models to downstream tasks with few labels while ensuring their reliability


### Originality
- I find the use of evidential learning for this setting to be original. The idea of preserving prior knowledge and increasing the evidence for all classes with evidential learning for this use-case is novel

- the strategy of inducing diversity of ensembles with different strengths for incorrect evidence regularization is novel for the evidential learning literature and has the benefit of not needing communication between ensemble particles during training -> improved computational efficiency

### Clarity
- the paper is overall clearly written conveying a well argued story about the behavior observed for different PEFT methods in the few-shot regime, then studying it with evidential learning lens and then proposing a few solutions to mitigate the encountered non-desirable behaviors

- the code comes to support information potentially missing from the paper

### Quality

- the authors conduct numerous experiments under several few-shot learning regimes, showing improved performances. They also conduct several ablations and sensitivity studies towards better understanding the impact of the building blocks of this method and the choices made for them

- code is provided in the supplementary for reproducibility of results

- the authors study their work on several PEFT methods from computer vision literature, mostly VPT in the main paper and the others in supplementary

**Weaknesses:**

### Framing of few-shot learning task
- the few-shot learning setting considered here is quite different from the one considered in most of the few-shot learning/meta-learning literature with many small sets of support images and classes and query images (sampled from miniImageNet, Omniglot, CIFAR, etc.), sometimes called _"episodes"_. In contrast the proposed few-shot learning setting proposed here keeps the original number of classes and reduces the number of samples to 1-20 samples per class. This ends up as a different setting as the model is finetuned on 200 to 4000 images for CIFAR100 for instance from the 1-shot to the 20-shot setting, which is not a big dataset but still not that few images to train on.

- I'm wondering whether the proposed setup of having many classes, but few samples, may be the source of the under-confidence behavior of PEFT methods for the few-shot learning setup that is proposed here. In such cases, simple solutions from the few-shot learning literature that do not require training: e.g., cosine classifier [b], [c]  (i.e., averaging features from the same class, do dot-product with query features, then softmax on similarities)

- the authors could refer to this setting that they propose by a different name to avoid confusion with few-shot learning where models can be finetuned with very few samples and there are several Bayesian methods there already. A potential alternative could be: low-shot regime.


### Motivation for evidential learning approach and few baselines
- the formalism based on evidential learning is interesting, but is not always trivial to implement and previous works [d],[e] reported difficulties in scaling evidential methods and the Dirichlet predictor to larger number of classes and encoders.

- while the approach itself is clearly described, the motivation for using such a line of approach instead of others that are more simple, generic and with a wider adoption, e.g., last-layer methods like Laplace-Redux [f], Bayesian last layer [g]. Since the endeavor of this work is to look at more practical settings with few labeled samples and reliability constraints, it may make sense to compare this work with some simple baselines to show the benefits of the approach and its trade-offs.
Other potentially relevant baselines would be the cosine classifier from few-shot learning, LoRA ensembles [h], linear probing with test-time augmentation [i]

- It would be also interesting to expand the list of PEFT methods studied with more recent and potentially more parameter efficient ones like LoRA [j] or VeRA [k] that have been shown useful to large transformer networks and could expand the scope of this work beyond image classification.


### Missing related works
- as mentioned above, the field of few-shot learning/meta-learning has seen several Bayesian methods with uncertainty quantification. Often these meta-learning methods fine-tune the network at test-time with just a few labeled samples (from classes not seen during pre-training). Here is a non-exhaustive list of relevant works from this area: PLATIPUS [m], VERSA [n], LLAMA [o], Bayesian MAML [p], Bayesian TAML [q], etc.

- a discussion on how the current B-PEFT is different from them and what type of problems it addressed and with which benefits would be useful here.

- the idea of converting a pre-trained network into a Bayesian Neural Network and training over a short period of a few epochs (not a low-shot setting though) has been recently proposed in ABNN [l]. The diversity-inducing mechanism with different incorrect belief regularization strengths can also be related with the random prior mechanism from ABNN [l] where an ensemble of networks was obtained from a pre-trained network and the diversity was induced by introducing a different class-level prior for each ensemble member. ABNN is pretty recent so it can be considered as concurrent work.


### Misc.
- the OOD experiment could benefit from the inclusion of typical metrics used in literature (e.g., AUROC, FPR95, AUPR, etc.) to allow easier comparison and putting the performance in context of other works.


### Minor - Novelty
- the evidential learning choice seems to be inspired to some extent by the multidimensional belief quantification work [37] that used uncertainty for vanilla few-shot learning / meta-learning. The authors however expand the idea further with belief strenghtening and incorrect belief regularization for ensembling.




**References:**

[a] Vinyals et al., Matching networks for one shot learning, NeurIPS 2016

[b] Gidaris et al., Dynamic Few-Shot Visual Learning without Forgetting, CVPR 2018

[c] Qi et al., Low-Shot Learning with Imprinted Weights, CVPR 2018

[d] Joo et al., Being Bayesian about Categorical Probability, ICML 2020

[e] Franchi et al., Encoding the latent posterior of Bayesian Neural Networks for uncertainty quantification, arXiv 2020

[f] Daxberger et al., Laplace Redux -- Effortless Bayesian Deep Learning, NeurIPS 2021

[g] Kristiadi et al., Being bayesian, even just a bit, fixes overconfidence in relu networks, ICML 2020

[h] Balabanov et al., Uncertainty quantification in fine-tuned LLMs using LoRA ensembles, arXiv 2024

[i] Ashukha et al. Pitfalls of In-Domain Uncertainty Estimation and Ensembling in Deep Learning, ICLR 2020

[j] Hu et al., Low-rank adaptation of large language model, ICLR 2022

[k] Kopiczko et al., VeRA: Vector-based Random Matrix Adaptation, ICLR 2024

[l] Franchi et al., Make Me a BNN: A Simple Strategy for Estimating Bayesian Uncertainty from Pre-trained Models, CVPR 2024

[m] Finn et al., Probabilistic Model-Agnostic Meta-Learning, NeurIPS 2018

[n] Gordon et al., Meta-Learning Probabilistic Inference For Prediction, ICLR 2019

[o] Grant et al., Recasting gradient-based meta-learning as hierarchical bayes, ICLR 2018

[p] Kim et al., Bayesian Model-Agnostic Meta-Learning, NeurIPS 2018

[q] Lee et al., Learning to Balance: Bayesian Meta-Learning for Imbalanced and Out-of-distribution Tasks, ICLR 2020

**Questions:**

This paper takes an interesting direction of study: ensuring reliability of PEFT foundation models in the low-shot learning regime.
I find the endeavor of the authors nice, with a good story and several PEFT methods. However I do have several concerns regarding the chosen few-shot learning setup (different from the common practices in the literature) and not necessarily realistic, the limited motivation brought for the use of evidential learning as well as the limited baseline. In addition, the relatively rich literature of Bayesian methods for few-shot learning is ignored.

My current rating is leaning towards reject at this time (a bit on the fence), but I'm looking forward for the rebuttal.

Here are a few questions and suggestions that could be potentially addressed in the rebuttal or in future versions of this work (please note that suggested experiments are not necessarily expected to be conducted for the rebuttal):

1. Please argue why using evidential learning should be used for this setting rather than last-layer or other Bayesian inspired methods

2. What is the performance of a cosine classifier in this setting? What about Laplace-Redux or LoRA ensembles?

3. How is this method working with LoRA-like methods, e.g., VeRA?

4. Why considering the proposed few-shot learning setup instead of the one from legacy few-shot learning / meta-learning literature

5. Can the authors discuss the difference and benefits of Bayesian-PEFT w.r.t. Bayesian few-shot learning methods?

6. Inclusion of OOD specific metrics in the OOD experiment.

**Limitations:**

The authors addressed some limitations in the supplementary.

---

> ### Author Rebuttal · Authors · 2024-08-07
>
> Thank you for taking the time to review our paper. We appreciate your constructive comments, thorough review, and suggestions. Here are the responses to the clarifying questions:
>
> **Q1: Framing of few-shot learning problem**
>
> In this work, we consider few-shot classification problem, i.e., the $N$-way $K$-shot classification problem, where the $N$-class classifier has $K$ examples per class to learn from. For instance, 1-shot Cifar100 is a 100-way 1-shot classification problem where the support set has a total of 100 examples (i.e., 1 example per class), and the model is evaluated on the test set (identical to the query set in the meta-testing tasks). We clarify that the current work does not rely on the episodic learning paradigm of meta-learning (matching networks, MAML, PLATIPUS), where both meta-training and meta-testing are done on task level in an episodic fashion with a large number of N-way K-shot tasks.  In fact, we consider more challenging few-shot learning tasks (e.g., 100-way 1-shot in Cifar100 and 102-way 1-shot in Flowers102) compared to the commonly used 5-way 1-shot meta-learning tasks. We leverage the power of the pre-trained foundation models eliminating the need of task based episodic meta-training. From a meta-learning perspective, the pre-training phase for the foundation model could be viewed as the meta-knowledge acquisition phase (i.e., meta-training). The pre-trained model can be seen as an expert equipped with meta-knowledge, and parameter-efficient fine-tuning technique performs quick adaptation to the downstream tasks, analogous to the support-set based adaptation done in meta-testing.  To test our method in the standard 5-way 1-shot tasks, we apply our model to the mini-ImageNet dataset test tasks and compare it with representative meta-learning models. The results are summarized in Table 1 in the attached PDF. The PEFT-based model outperforms the episodic training meta-learning models demonstrating the potential of large vision foundation models for effective few-shot learning.
>
> **Q2: Number of classes is the source of Under-confidence**
>
> Please refer to the answer to **Q1** in the overall response.
>
> **Q3: Performance of simple solutions that do not require training**
>
> As suggested by the reviewer, we carry out experiments with cosine classifier without training for the 100-way 1-shot task on Cifar100. The results are shown in Table 6 of the PDF file. The cosine classifier has comparable generalization performance (accuracy) in comparison to VPT based model. However, looking at the ECE, the miscalibration issue is even higher than VPT based model. Hence, the simple solution (cosine classifier) does not ensure calibrated predictions.
>
> **Q4: Low-shot regime suggestion**
>
> Thank you for the suggestion. We will add a detailed discussion of the few-shot learning setup, clearly explaining that we differ from  the existing meta-learning works as we no longer rely on the meta-training phase and episodic learning paradigm.
>
> **Q5: Scaling evidential methods and the Dirichlet predictor to larger number of classes and encoders.**
>
> The lack of scalability of EDL has been observed and addressed by Ref [38] (Learn to Accumulate Evidence from All Training Samples: Theory and Practice, ICML 2023) that enables the evidential models to be scaled to ResNet for Cifar100, and we further extend the evidential framework to Vision Transformers in challenging 100-way, 101-way and 102-way few-shot tasks further demonstrating the scalability of evidential models.
>
> **Q6: Comparison with baselines, e.g., last-layer methods like Laplace-Redux, Bayesian last layer, and linear probing with test-time augmentation**
>
> We carry out experiments with the suggested models on 100-way 1-shot Cifar100 dataset. The results are presented in Table 6. We observe that these methods also suffer from the under-confidence issue when straightforwardly extended to the VPT. We resort to the Evidential Deep Learning framework to address the under-confidence issue of the VPT model. Similarly, we also fine-tune the ViT model using Lora as another parameter efficient fine-tuning method. The fine-tuning results in under-confidence issue as well. Further expanding the scope of the work to tasks beyond image classification with LoRA and VeRA based methods can be an interesting followup work. Thank you for the suggestion.
>
> **Q7: Discuss the difference and benefits of Bayesian-PEFT w.r.t. Bayesian few-shot learning methods?**
>
> Please refer to the answer to **Q2** in the general response for a detailed discussion of the proposed method over Bayesian inspired models, which include all the Bayesian few-shot learning methods. As a result, these Bayesian models are inadequate to address the unique under-confidence behavior associated with the PEFT methods. Furthermore, benefiting from a large pre-trained vison foundation model, the proposed method is able to achieve much higher prediction performance as compared with the Bayesian few-shot learning methods trained through episodic meta-training with the specifically constructed few-shot tasks. In Table 1 of the attached PDF file, we show the results on 5-way 1-shot mini-ImageNet. As can be seen, our model outperforms the bayesian few-shot learning models in existing benchmark task by a large margin.
>
> **Q8: ABNN [l] Discussion**
>
> ABNN introduces bayesian normalization layers after training of deep learning model, and requires additional training of these layers in a post-hoc manner. In theory, an evidential deep learning model could be augmented with the bayesian normalization layers/ABNN idea as an alternative to belief-based diversity. We will include discussion of ABNN in the updated draft, and leave exploration of ABNN based diversity for bayesian evidential model as a potential future work.
>
> **Q9: Inclusion of OOD specific metrics in the OOD experiment.**
>
> Please see the answer to **Q3** in the overall response.

---

> > ### Comment · Reviewer_7WZR · 2024-08-10
> > **Post-rebuttal comments**
> >
> > I would like to thank the authors for the detailed and informative rebuttal. I imagine they invested a lot of effort into clarifying the concerns from the 4 reviewers and I'm confident they will improve this work.
> >
> > I've read the other reviews and the responses of the authors to them and skimmed through the paper again.
> >
> > The strong points I see in this rebuttal are:
> > - several clarifications on the motivation of using evidential models for this tasks and observed under-confidence phenomenon, clarification on few-shot learning setting
> > - new studies to better understand the under-confidence trend w.r.t. prompt size, number of tuned parameters, dataset size
> > - new results comparing with simple baselines that don't need training at all (cosine classifier, test-time adaptation) or limited training (LoRA, Laplace Redux) highlighting the effectiveness of this method but also the presence of under-confidence or low calibration to them
> > - results on typical few-shot learning/meta-learning protocols on mini-Imagenet
> > - reporting of typical OOD metrics in OOD experiments
> >
> > A few minor things that could be improved:
> > - for the OOD experiments (Table 4 in the submitted PDF), only the performance of the proposed model is shown. It would be useful to report the performance of the vanilla PEFT method
> > - the description of the few-shot protocol for the CIFAR100 results could be added (number of sampled episodes in the evaluation)
> >
> >
> > Overall, I think the authors did a nice job in the rebuttal and provided convincing responses.
> > I will raise my rating to 6 - Weak Accept.

---

> > > ### Author Response · Authors · 2024-08-11
> > > **Official Comment by Authors**
> > >
> > > Dear Reviewer 7WZR,
> > >
> > > Many thanks for carefully checking our rebuttal. We are happy that it has adequately addressed the main concerns and we will make sure to incorporate these changes into the revised paper. We also appreciate that the reviewer increases the rating to 6!
> > >
> > > We would like to take this opportunity to address the two additional inquiries from the reviewer. First, in the table below, we present the performance of the vanilla PEFT method for the OOD experiments, which is worse than ours.
> > >
> > > |     Setting    | Method | AUROC | AUPR | FPR95 |
> > > | ------------- | --------- | --------- | ------- | -------- |
> > > | 100 way 1 shot |  PEFT  | 79.53 | 80.09| 70.58 |
> > > | 100 way 1 shot | B-PEFT | 81.24 | 81.98| 68.15 |
> > > | 100 way 5 shot |  PEFT  | 90.93 | 90.75| 39.82 |
> > > | 100 way 5 shot | B-PEFT | 92.58 | 92.85| 35.24 |
> > >
> > > Second, for the few-shot protocol on CIFAR100, we have 3 settings. For 5-way 1-shot and 10-way 1-shot, we formulate a task by randomly selecting 5 and 10 classes, respectively. In each such task, the support set contains 1 sample and the query set contains 50 samples for each of the selected classes. For 100-way 1-shot, we formulate a task by randomly selecting 1 sample per class in the support set. The query set contains $100$ samples per class. The result for each setting is obtained by averaging the performance across 50 tasks.

---

> > > > ### Author Response · Authors · 2024-08-13
> > > > **A kind reminder**
> > > >
> > > > Dear Reviewer 7WZR,
> > > >
> > > > We noticed that the updated score has not been reflected in openreview yet. Since the discussion deadline is approaching, we are grateful if you could make this update soon. Many thanks!

---

### Author Rebuttal · Authors · 2024-08-07

## Overall Response

We thank all reviewers for their valuable feedback and constructive suggestions. We identify some important questions raised by multiple reviewers and answer them together in our general response below.

**Q1: Under-confidence behavior of PEFT methods w.r.t. number of classes (Reviewer 7WZR), data size (Reviewers 7WZR and CbTK), and number of unfrozen parameters (Reviewer 7WZR).**

In order to more thoroughly understand the under-confidence behavior of PEFT methods under diverse settings as suggested by the reviewers, we refer to some of already reported results in the paper while conducting additional experiments. The new results are presented in the attached PDF file. We summarize our main findings as follows:

- *Data size*: In Table 1 of the paper, we varied the number of training samples per class from 1 to 20. We observe  that the model's accuracy increases with more training samples. However, under-confidence remains, even with the increase in training samples. We conduct additional experiments on Cifar100 by increasing the training samples per class to 500  and observe the under-confidence issue despite the increase in the accuracy. The trend is summarized in Figure 2 (c-d) of the attached PDF. We see an increase in accuracy and a decrease in ECE. However, even with 500 samples per class, the under-confidence issue remains. Further, we report the accuracy and ECE of the fully fine-tuned model (fine-tuning of all the parameters) for 1 shot cifar100 in Table 5 of the attached pdf. We observe a decrease in accuracy. The miscalibration issue still remains. We observe the overconfidence behavior by looking at the reliability plot as presented in Figure 1 (a). The performance suggests the overfitting of the fully tuned model.


- *Number of classes*: To study the relationship between model accuracy, uncertainty, and the number of classes, we formulate standard 5-way 1-shot MiniImagenet tasks and apply the PEFT to these tasks. The results are summarized in Table 2 of the attached PDF. As can be seen, the model achieves high overall accuracy of 89.78\%. However, the model remains under-confident with an ECE of 0.418, where under-confidence is shown by the the reliability diagram in Figure 1. We further formulate 5-way 1 shot, 10-way 1 shot, and 100-way 1 shot tasks in Cifar100. The results are presented in Table 3. As we decrease the number of shots from 100 to 5, we see an increase in accuracy and a decrease in ECE. However, the under-confidence remains.

- *Number of unfrozen parameters*: We conduct additional experiments on Cifar100 via 100-way 1-shot tasks by varying the number of prompts for 1) shallow prompt: prompt added to the input only and 2) deep prompt: prompt added to all Transformer encoder layers' input as well. The accuracy and ECE trends are presented in Figure 2 (a)/(b) of the pdf. As can be seen, with the increase in the number of prompts for both shallow and deep prompts, there are fluctuations in accuracy and ECE performance. However, the under-confidence issue persists for all the cases.

**Q2: Justification of using evidential learning to achieve calibrated PEFT (Reviewers 7WZR and  YJfL)**

As compared with the Bayesian-inspired models, evidential learning offers two key properties that allow us to formulate a principled solution to address the unique under-confident behavior of the PEFT methods. First, thanks to its evidence-based fine-grained uncertainty decomposition capability, we can separate two distinct sources of second-order uncertainty, including vacuity (line 172) and dissonance (line 176). Different from the commonly used first-order uncertainty (e.g., entropy), these two second-order uncertainty serve as a key tool to understand why PEFT methods are both accurate (with a low dissonance) while being under-confident (with a high vacuity). This key insight suggests that these methods systematically under-estimates the contribution from the prior knowledge to the downstream task. While the classical Bayes’ theorem offers a principal idea to address the issue, which is to strengthen the prior belief, there is a lack of practical way to achieve this. As the second key property, evidential learning allows us to leverage the base rate, which is rooted in the subjective logic theory as an effective vehicle to adjust the prior belief gained through pre-training. To this end, we propose a transformation function in Eq (6) to adjust the base rate that leads to the increase of the model confidence while maintaining the predictive accuracy of the model as guaranteed by our theoretical results in Lemma 2 and Theorem 3.

 We also conduct additional experiments and compare our approach with Bayesian inspired models for 100-way 1-shot Cifar100 tasks. The results are reported in Table 6.  We use laplace approximation on last layer of the model using Kronecker Product and Diagonalization represented by KronLaplace and DiagLaplace in the table. As can be seen, the Bayesian-inspired models do not address the under-confidence issue, which still exhibits a fairly high ECE. This further confirms the effectiveness of the proposed evidential learning-based approach.

**Q3:  OOD experiments and Clarification (Reviewers 7WZR and Dmeg)**

 We present the OOD results of Cifar10 as in-distribution dataset and Cifar100 as out-of-distribution dataset with AUROC, FPR95, AUPR metrics for our model on $100$-way $1$-shot and $100$-way $5$-shot cifar100 tasks in Table 4. As seen, with more training data, the model's ood detection capabilities improve. Even with only 5 samples/class (i.e. $100$-way $5$-shot cifar100 task), the model can achieve AUROC of 92.58.

---

### Decision · Program_Chairs · 2024-09-25

**Decision:**

Accept (poster)

**Comment:**

The paper proposes a Bayesian Parameter Efficient Fine-Tuning framework for training large transformer-based foundation models and reducing underconfidence while ensuring reliable prediction in few-shot settings. The reviewers were broadly positive about the paper. They highlighted that they were generally convinced by the empirical evaluation presented in the paper and agreed that the problem of underconfidence due to PEFT is interesting and important. While no reviewer strongly recommends acceptance, I agree that the problem studied in the paper and the proposed solution are of interest to the NeurIPS community and that the paper should be presented at the conference.